# Fully printed zero-static power MoS$_2$ switch coded reconfigurable graphene metasurface for RF/microwave electromagnetic wave manipulation and control

Xiaoyu Xiao [1,7], Zixing Peng[2,7], Zirui Zhang[1], Xinyao Zhou[1], Xuzhao Liu[3], Yang Liu[3], Jingjing Wang[2], Haiyu Li[1], Kostya S. Novoselov [4,5,6], Cinzia Casiraghi [2] & Zhirun Hu [1,5] ✉

Reduction of power consumption is the key target for modern electronic devices. To this end, a lot of attention is paid to zero-static power switches, being able to change their state between highly resistive and highly conductive and remain in this state even in the absence of external voltage. Still, the implementation of such switches is slow because of compatibility issues of new materials with CMOS technology. At the same time, printable technology enables low-cost processes at ambient temperature and integration of devices onto flexible substrates. Here we demonstrate that printed Ag/MoS$_2$/Ag heterostructures can be used as zero-static power switches in radiofrequency/microwave spectrum and fully-integrated reconfigurable metasurfaces. Combined with graphene, our printed platform enables reconfigurable metasurface for electromagnetic wave manipulation and control for wireless communications, sensing, and holography. In addition, it is also demonstrated that the localised MoS$_2$ phase change may have promoted Ag diffusion in forming conductive filaments.

In the modern information-driven era, smart wireless connectivity has become an essential part of our daily life. Such smart connectivity has traditionally relied on electrically reconfigurable wireless communication systems[1–3]. Most of today's reconfigurable technologies are based on semiconductor switches (such as transistors, PIN diodes, light guide elements, etc.). Recently, liquid crystals and phase change memory materials, such as vanadium dioxide, have been proposed for realization of system reconfigurability[4,5]. Still, all these switches have a common limitation: they require a DC holding voltage and dissipate a considerable static power for their operation. On the other hand,

zero-static power switches (also known as non-volatile switches, or memristive switches) can greatly increase the energy efficiency for reconfigurable systems, especially when a reconfigurable system has a large number of switches, as such devices do not dissipate static power. In recent years, researchers have integrated zero-static power switches to radiofrequency (RF) and microwave antenna and circuit applications[6–9].

2-dimensional (2D) materials are very attractive for non-volatile switches because they enable simple device structure and fabrication process as compared to other types of RF switches, as well as low

[1]Department of Electrical and Electronics, University of Manchester, Manchester M13 9PL, UK. [2]Department of Chemistry, University of Manchester M13 9PL, Manchester, UK. [3]Department of Materials, University of Manchester M13 9PL, Manchester, UK. [4]Department of Physics and Astronomy, University of Manchester, M13 9PL, Manchester, UK. [5]National Graphene Institute, University of Manchester, M13 9PL Manchester, UK. [6]Institute for Functional Intelligent Materials, National University of Singapore, Singapore 117544, Singapore. [7]These authors contributed equally: Xiaoyu Xiao, Zixing Peng. ✉e-mail: z.hu@manchester.ac.uk

programming voltages and nanoscale dimensions[10–12]. Several efforts have been made to investigate 2D material enabled zero-static power RF/microwave switches[6,7,10–16], for example high frequency zero static power mono- and bi-layer $MoS_2$ switches have been demonstrated[12]. However, a fully printed platform based on a reconfigurable meta-surface for electromagnetic wave manipulation and control, and capable of providing the desired propagation modes for wireless communications, sensing, and holography, has yet to be reported.

Here we demonstrate a fully printed $MoS_2$ zero-static power switch integrated graphene metasurface. These metasurfaces are flex-ible, inexpensive, and biodegradable, addressing not only the low-cost applications for the Internet of Things (IoTs) but also sustainability-driven reduction of e-waste, while maintaining a high performance[17–19]. However, all the reported printed graphene antennae/metasurfaces are passive and cannot be reconfigured, which hinders their applications where reconfigurability, such as beamforming and directional sensing, is required. In this work, we show a fully printed technology based on a low-cost, environmentally friendly, stable, and highly conductive gra-phene ink and water-based inkjet printable $MoS_2$ ink[20]. The zero-static power $MoS_2$ switches are integrated with the graphene metasurface to provide non-volatile reconfigurability that can switch between several radiation patterns through 1-bit coding.

In contrast to previous works based on the use of commercially available transistors and diodes, made with traditional (rigid) semi-conductors, we report a fully printed zero-static power reconfigurable graphene metasurface on a flexible substrate that is suitable for low-cost and disposable IoT applications.

## Results and discussion
### Inkjet-printed zero-static power RF/microwave $MoS_2$ switches

Two types of $MoS_2$ switches were inkjet-printed: crossbar layout (active area of $0.2 \times 0.2$ mm$^2$) and microstrip layout (active area of $0.05 \times 0.05$ mm$^2$). The former is designed for integration with the screen-printed metasurface working in lower GHz as it enables easy alignment with the screen-printed patterns, whereas the latter is used for RF/microwave performance evaluation. The cross-section view of an inkjet-printed $Ag/MoS_2/Ag$ heterostructure measured by High-angle annular dark-field scanning transmission electron microscopy (HAADF-STEM) is shown in Fig. S1. There are clear boundaries between Ag, $MoS_2$ and the paper, used as substrate. The nanosheets arrangement in the printed film is relatively ordered, as compared to other methods such as vacuum filtration, however some porosity and random arrangement is typically observed in printed films made of nanosheets[21]. The thickness of the $MoS_2$ layer is ~2.2 µm (Fig. S1). Note that thinner films have shown shortcuts between the top and bottom Ag electrodes (Fig. S2), hence this thickness was selected.

An optical picture of an inkjet-printed crossbar $Ag/MoS_2/Ag$ switch is depicted in Fig. 1a. Figure 1b shows the typical non-volatile I-V characteristics of an $Ag/MoS_2/Ag$ switch measured over 300 cycles, demonstrating stable on/off-resistance and relatively small set voltage, comparable with values measured for most of the reported printed memristive switches (also known as memristors) (Table SI). At the outset, a $MoS_2$ switch is typically off-state (State 0) until the applica-tion of a voltage to turn the device to on-state (State 1). Then the switch persists in the on-state until a negative bias resets it to off-state. During the measurement, a forward current limit of 11 mA is preset to reduce possible damage to the device in the case of non-volatility. The highest set voltage is around 1.75 V and highest reset voltage around −1.1 V, respectively. The endurance of the inkjet-printed $MoS_2$ switch is shown in Fig. 1c, where the device has been switched on/off manually up to 300 cycles continuously. It can be derived from the figure that the average on-resistance is about 10 Ω, which is the lowest on-state resistance among all printed inorganic memristive switches so far

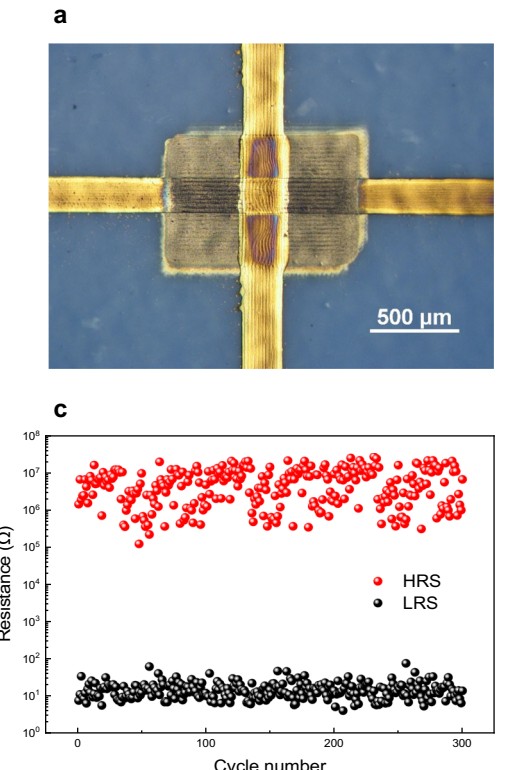

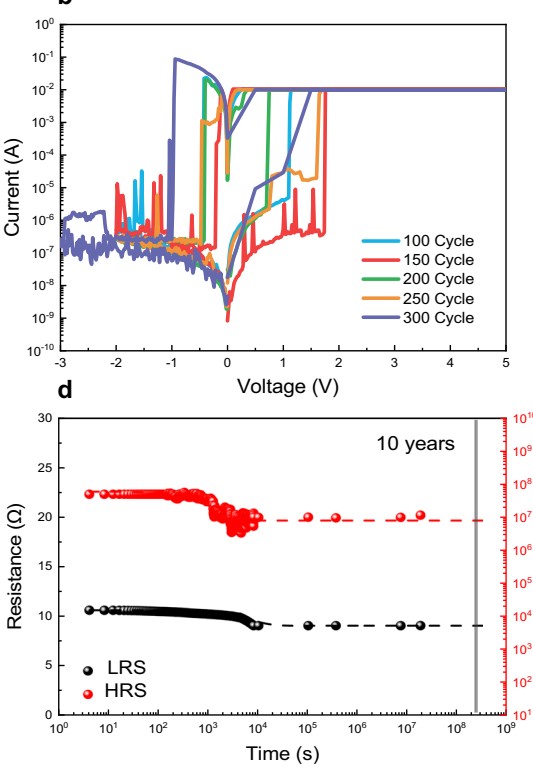

**Fig. 1 | DC characteristics of the zero-static power $Ag/MoS_2/Ag$ switch. a** Top view of an inkjet-printed $MoS_2$ switch (active area of $0.2 \times 0.2$ mm$^2$). **b** Measured I–V characteristics of the switch. **c** Endurance (resistance distribution) of the switch with 300 manual DC switching cycles. **d** Retention of the switch, measured over

$10^7$ s at room temperature. An exponential extrapolation of the data (dashed lines) suggests that the device switching on/off ratio will retain average of $6 \times 10^5$ for over 10 years.

(Table SI). The average on/off-ratio is around $6 \times 10^5$ with average off-resistance of $6 \times 10^6$ Ω. Overall, the printed MoS$_2$ memristive switch shows performance comparable to the best printed switches reported in literature (Table SI, and Fig. S3); to note that previous devices were printed on silicon or plastic, while in our case the device is printed on paper.

The retention of the device was measured at room temperature every 4 s up to $1.8 \times 10^7$ seconds (about 7 months), as shown in Fig. 1d. For the on-state measurement, a small bias voltage of 0.05 V was set. The device operates stably for both on- and off-states.

Photos of the inkjet-printed microstrip MoS$_2$ memristive switch are illustrated in Fig. 2a The switch was printed directly on a purposely designed microstrip line on Rogers 5880 substrate. The equivalent circuit of the MoS$_2$ memristive switch at RF/microwave was modelled by a resistor and capacitor, shown in Fig. 2b. The RF/microwave properties of the switch have been investigated by measuring the device's scattering parameters (S-parameters). The intrinsic S-parameters of the switch are de-embedded and depicted in Fig. 2c, d (The de-embedding details are given in S1.3). The intrinsic RF characteristics of the MoS$_2$ switch, such as the RF/microwave on-, off-resistance ($R_{on}$, $R_{off}$), off-capacitance ($C_{off}$), and figure of merit (FoM) have been extracted from the device intrinsic S-parameters based on the equivalent circuit model: $R_{on} = 8.9$ Ω, $R_{off} = 1.4$ MΩ and $C_{off} = 31.92$ fF. The $FOM = (1/(2\pi R_{on}C_{off}))$ of the switch is 0.56 THz. The switch has insertion loss <0.7 dB at the on-state and better than 10 dB at the off-state from DC to 12 GHz.

## Zero-static power MoS$_2$ switching mechanism

Figure 3a shows the Raman spectra of an inkjet-printed MoS$_2$ film excited by a laser at 514.5 nm wavelength. The characteristc $E^1_{2g}$ mode at ~383 cm$^{-1}$ and $A_{1g}$ mode at ~408 cm$^{-1}$ of 2H MoS$_2$ can be observed[22]. Besides, the 2H-MoS$_2$ crystal structure of the pristine sample can be confirmed by the HAADF-STEM image by fast-Fourier-transform (FFT) shown in Fig. 3b. The selected area electron diffraction (SAED) images of this MoS$_2$ layer are corresponding to [3 4 −1] and [1 0 3] lattice faces. The atomic resolution HAADF-STEM image of distorted MoS$_2$ flakes in Fig. 3d clearly indicates that the positions of the S and the Mo atoms (see arrows) agree with the lattice structure of 2H phase MoS$_2$.

To further investigate the mechanism of conductive filament formation, the changes in the Ag content were measured by comparing a pristine sample to a used sample (switched on/off 20 times, the sample was at on-state, i.e., LRS). It is found that the Ag content increases from 0 at.% to 0.3 at.% in the middle part of the MoS$_2$ layer, as depicted in Fig. 3c, indicating Ag diffusion and migration into the MoS$_2$ layer upon application of the external electric field. To note that the middle part of the MoS$_2$ layer was selected on purpose to avoid artefacts due to Ag that may have already migrated into MoS$_2$ layer from the Ag/MoS$_2$ interface during the inkjet printing process, i.e., before the bias is applied to the device. This result further confirms that the device switching mechanism is based on electro-migration[23,24].

However, the conventional conductive filament formation mechanism, which has been observed also in metal oxide memristors[25], might not work as well in printed layers of nanosheets: it

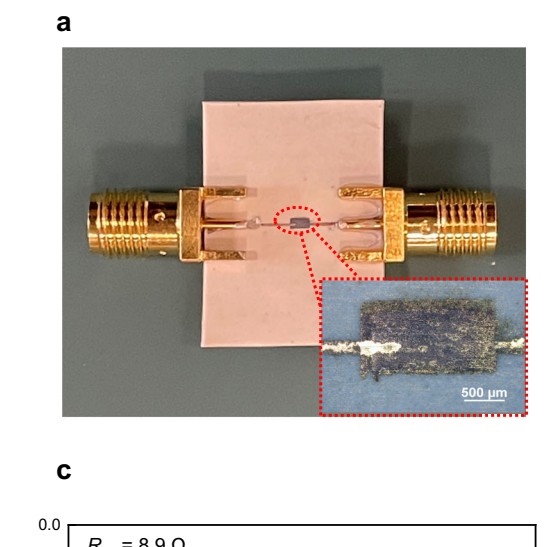

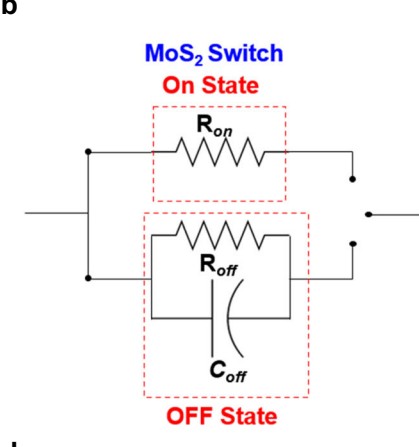

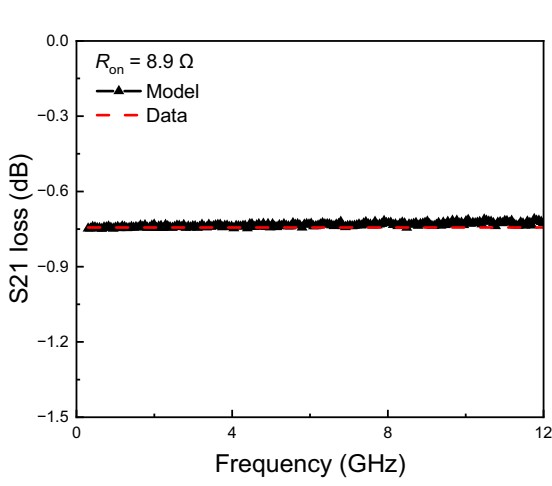

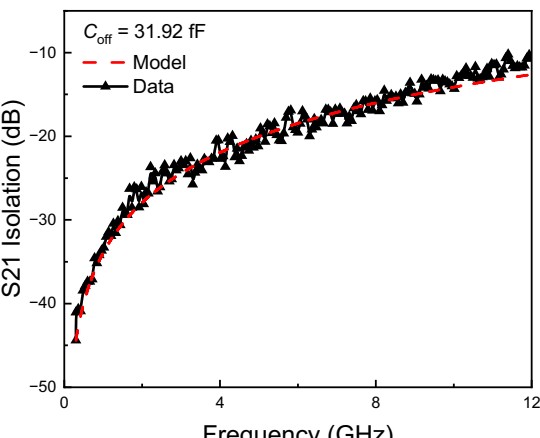

**Fig. 2 | RF properties of zero-static power Ag/MoS$_2$/Ag switch. a** Photo of the microstrip inkjet-printed RF/microwave MoS$_2$ memristive switch with SMA and enlarged MoS$_2$ switch (active area of $0.05 \times 0.05$ mm$^2$). **b** RF/microwave equivalent circuit model of the switch. **c, d** Measured de-embedded transmission coefficient S$_{21}$ for on- and off-state from 0. 3 – 12 GHz, respectively.

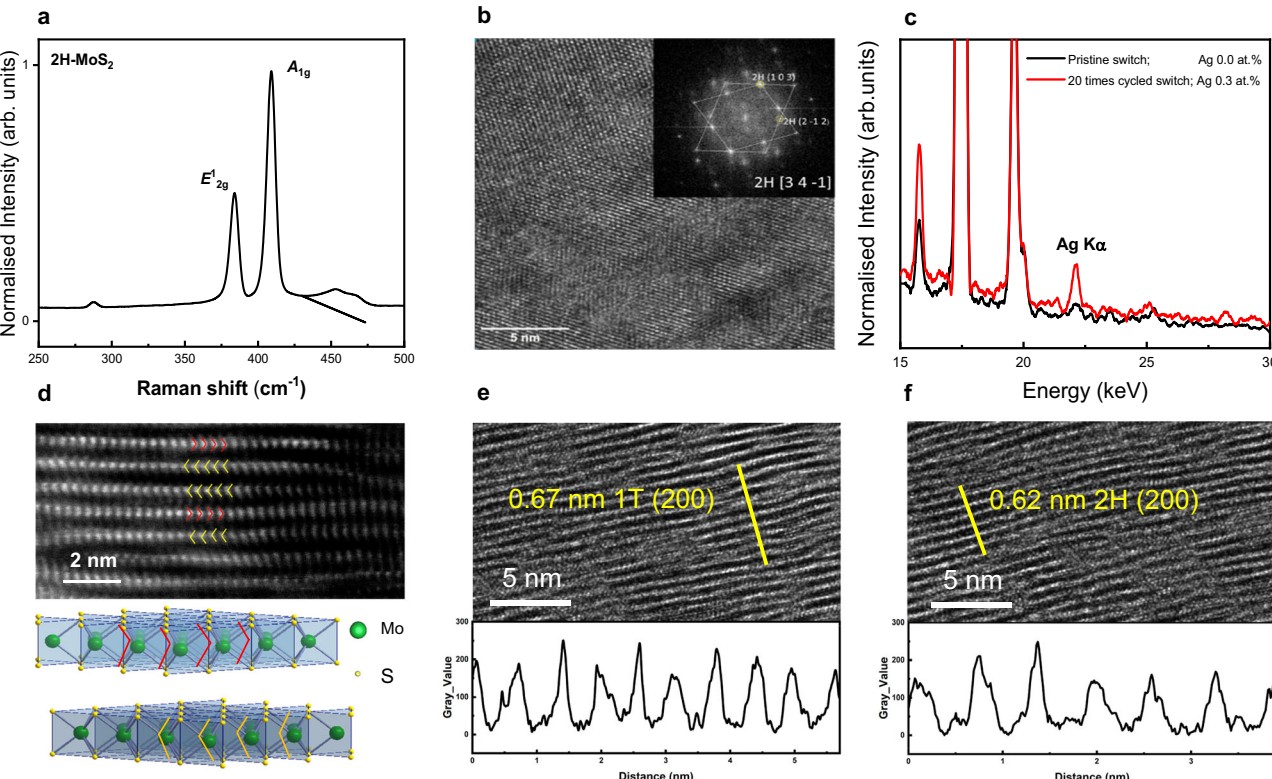

**Fig. 3 | HAADF images of zero-static power Ag/MoS₂/Ag switch. a** Raman spectra of 2H-MoS₂ sample. **b** Local phases of 2H-MoS₂ at a high resolution TEM image of the unused MoS₂ switch sample at 2H [3 4 −1] axis with FFT. The crystal is rotated or twisted along [3 4 −1] axis and two set of diffraction patterns share 2H (1 0 3) diffraction disc (inset). **c** EDX Ag $K\alpha$ region spectra acquired from 500 nm by 500 nm regions in the middle of unused and used samples, showing 0.3 at.% difference of Ag concentration. **d** Atomic resolution HAADF-STEM image and schematic of 2H phase MoS₂ crystal structure. **e, f** Regions with dislocation paths cross the MoS₂ at higher magnification (used sample), showing different lattice parameters which may indicate the formation of localised 1T-MoS₂ phase.

is well known that single layer 2D materials are difficult for ions to penetrate due to the high energy barrier caused in dense structure[26]. On the other hand, 2H MoS₂ is known to change its phase into the metallic 1T phase upon external stimuli such as ion intercalation and strain[27,28]. Therefore, HAADF-STEM on the pristine and used sample, Fig. 3e, f, was conducted. Different lattice parameters for the MoS₂ layers can been observed, indicating the formation of localised 1T-MoS₂ phase. Although reactions between Ag and bulk MoS₂ is not thermodynamically favourable and no reactions can be observed at the interface by X-Ray Photoelectron spectroscopy (XPS) measurements[29], the MoS₂ nanosheets show intrinsic defects including lattice bending, stacking faults and expansion of the MoS₂ layer structure after the voltage is applied (Fig. S12a, b), which can generate strain in the MoS₂ flakes (Fig. S13), leading to formation of localised 1T-MoS₂ phase[30] and to the variance in the electric properties[31]. Our measurements reveal that while a complete phase change in MoS₂ (from 2H to 1T) is highly unlikely, some localized phase change due to applied electric field as well as intrinsic defects in the solution-processed MoS₂ nanosheets may have catalysed the formation of Ag conductive paths to enable the device to show macroscopic conductivity at on-state.

**Proof-of-concept demonstration of fully printed zero-static power MoS₂ switch coded RF/microwave reconfigurable graphene metasurface**

Figure 4 shows the printing process of the MoS₂ switch coded RF/microwave reconfigurable metasurface, which consists of 36 inkjet-printed MoS₂ switches, 6 × 6 screen-printed graphene patches, paper/silicone substrate and ground plate. Paper substrate was used in this work for the purpose of better adhesion (to graphene ink), flexibility

and low-cost. It has a dielectric constant of 2.3, relative permeability of 1, the thickness of 0.1 mm, and the loss tangent of 0.02 at room temperature[17]. The flexible silicone layer (Polymax, SILONA transparent Silicone Sheet GP) with a dielectric constant of 2.9 has a thickness of 6 mm. The metal foil is set to eliminate electromagnetic waves transmitting through the metasurface as only reflection wave is considered. The graphene ink is made by liquid phase exfoliation using recycled Cyrene[32]. The MoS₂ switches are inkjet-printed on the top of the screen-printed graphene patterns, operating as active phase-tuning devices. Narrow graphene strips (screen-printed) placed along the graphene patch are DC bias lines to supply required DC voltages (to switch on/off the MoS₂ switches) via silver electrodes. Geometric details of the metasurface and its unit cell are given in Fig. S18 and Table SII.

In the full wave electromagnetic simulation (CST), the printed graphene was modelled as ohmic sheet which was set to have sheet resistance of 1.5 Ωsq⁻¹ based on the measurement using a four-point probe station (Jandel, RM3000). The unit cell has two separate parts connected by a switch (Fig. 5b, c and Fig. S18) and was designed to provide a 180° phase difference between on (two parts are connected) and off (two parts are disconnected) for a vertically incident plane wave as shown in Fig. 5a. The states of the unit cell are controlled by a non-volatile MoS₂ switch. In this work, a 180° phase shift was designed at 3.54 GHz, i.e., the metasurface operates at 3.54 GHz.

The photographs of the fully printed zero-static power MoS₂ switch coded RF/microwave reconfigurable graphene metasurface and its unit cell are illustrated in Fig. 5b, c, respectively. Figure 5d shows the far field pattern measurement setup in the anechoic chamber. To demonstrate the reconfigurability, the reflected waves were investigated under different coding with the incident angle fixed (in this

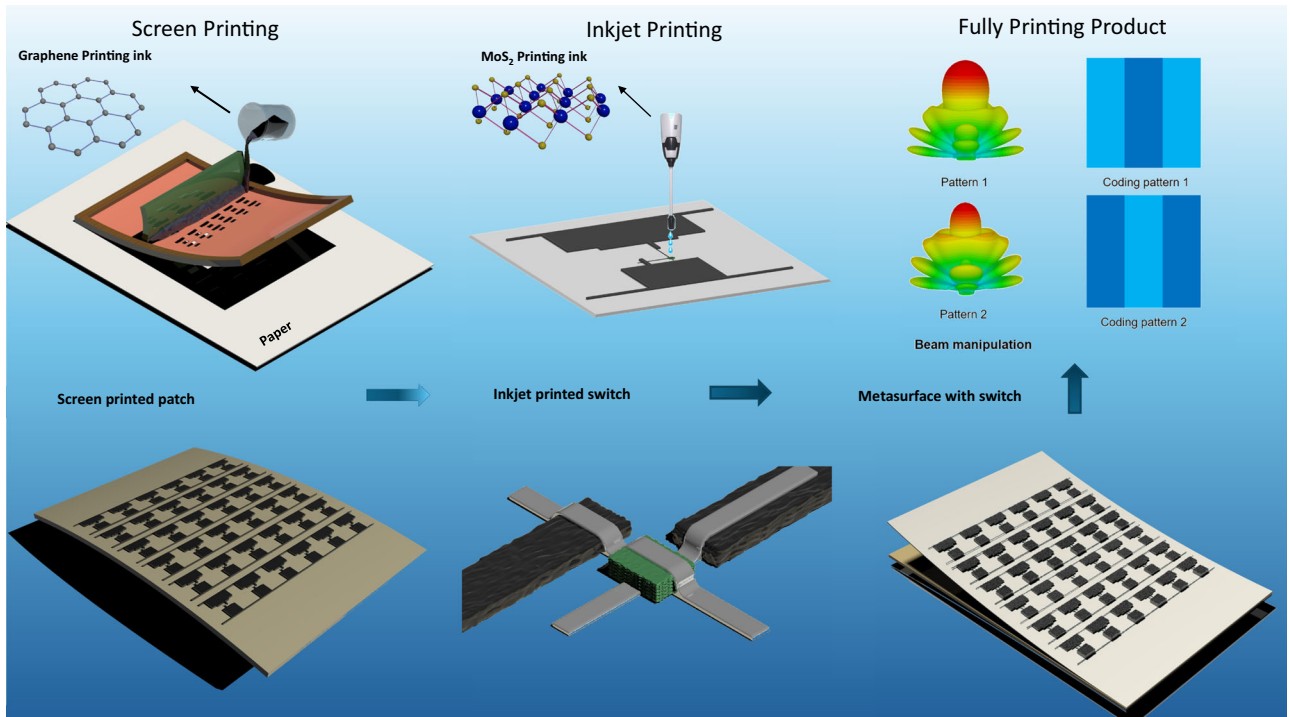

**Fig. 4 | Fully printed zero-static power MoS₂ switch coded reconfigurable graphene metasurface fabrication process.** The metasurface is printed on flexible paper with graphene ink through screen printing technology. The MoS₂ switches are then directly inkjet-printed on the top of the screen-printed graphene elements. The switch has MoS₂ as an active layer which is sandwiched by two silver electrodes. The fully printed reconfigurable metasurface is placed on a flexible silicone which is then adhered to a metal foil as a ground plate. Under different coding sequences, the metasurface can generate various propagation modes.

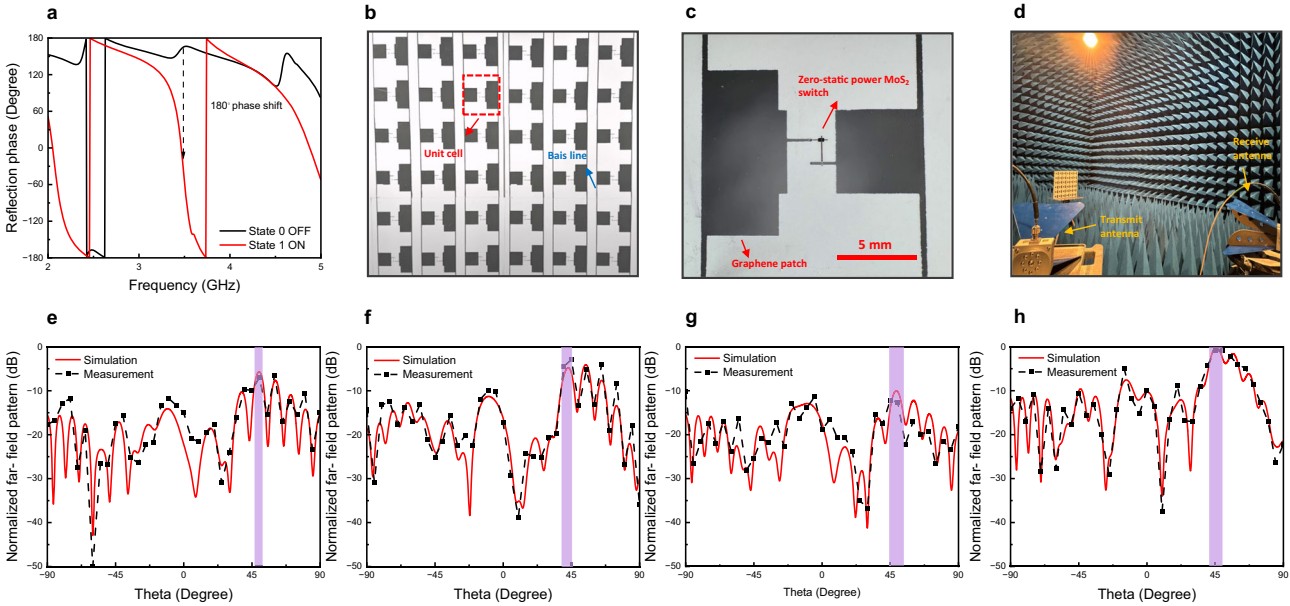

**Fig. 5 | Zero-static power MoS₂ switch coded RF/microwave reconfigurable metasurface and its performance. a** Simulated phase responses of the metasurface for a normal electromagnetic wave incidence. **b**, **c** Photos of the whole fully printed 6 × 6 zero-static power MoS₂ switch coded RF/microwave reconfigurable graphene metasurface and its unit cell, respectively. **d** Photo of the setup in the anechoic chamber. Measured far field patterns with coding sequence of (**e**) 001001, (**f**) 010010, (**g**) 101101, and (**h**) 001011 with an oblique incident angle of −50°.

work, the incident angle of − 50° was chosen arbitrarily), as shown in Fig. S19. The far field radiation patterns under different coding sequences were measured. It can be observed that the coding sequence '001001' in Fig. 5e provides very different reflected beam patterns compared to those of the coding sequences '010010' and '001011' in Fig. 5f and h, respectively. For the same incident angle, the main reflected beam in Fig. 5f points to 41° direction and in Fig. 5h point to 45° direction, whereas the main reflected beam in Fig. 5e directs to 52°, demonstrating the ability of the fully printed MoS₂ switch coded metasurface to designate a reflected wave to a desired

direction. This is highly desirable for smart wireless environment where intelligent reflective surfaces (IRS) can be deployed to make the wireless signal path programmable. If the coding sequence becomes '101101', the reflected electromagnetic wave is spread out into many beams as illustrated in Fig. 5g, revealing that the $MoS_2$ switch coded metasurface can manipulate the incident wave so to level the reflected wavefronts for RCS reduction. In addition to wireless communications and RCS reduction, the ability of $MoS_2$ switch coded metasurface to control and manipulate the reflected wavefronts can also be very useful for wireless sensing and microwave holography. Importantly, in all these applications, hundreds or thousands of switches are likely to be needed. The deployment of zero-static power switches will significantly improve the energy efficiency of the system as the switches dissipate zero-static power. In Table SIII, we have compared our zero-static power reconfigurable metasurface with those recently published works. The distinguished feature of the devices reported in this work is that they consume no static power, which is particularly desirable for all today's electrically reconfigurable systems, which need to be energy efficient to fit the net zero requirements. For instance, whether it is IRS or large reconfigurable antenna arrays in wireless communication systems, they require a large number of switching devices such as PIN diodes. The $MoS_2$ memristive switches demonstrated in this work have the potential to significantly reduce the systems' power consumptions.

This work has presented a fully printed zero-static power (also known as non-volatile, or memristive) $MoS_2$ coded RF/microwave reconfigurable graphene metasurface on a flexible substrate. The metasurface consists of 36 zero-static power $Ag/MoS_2/Ag$ switches and 36 graphene patches. By varying the state of the switches, each graphene patch can achieve 180°phase shift. The stable DC and RF characteristics of the $MoS_2$ switches prove that they can be applied for RF/microwave reconfigurability. Although the set on voltage of the $MoS_2$ switches in this work is relatively high (about 1.75 V) compared with most conventional transistor and diode switches, the advantage of the fully printed $MoS_2$ switch enabled active metasurface is that not only the active metasurface dissipates no static power, whereby significantly reducing the energy consumption especially in those applications that large number of switches are needed, but also the platform is fully printed, which is of great benefit for enabling low-cost additive manufacturing. Furthermore, the set on voltage can be reduced and optimized by controlling the thickness of the $MoS_2$ layer and the switch structure. Finally, it is demonstrated that the fully printed prototype zero-static power $MoS_2$ reconfigurable graphene metasurface can be electronically reconfigured to provide desired reflected wavefronts, which are highly desirable for programmable wireless environment, intelligent wireless sensing systems and microwave holography. With advancement of device structure and manufacturing process optimization, inkjet-printed $MoS_2$ switches incorporated with screen-printed graphene antennae, antenna arrays and metasurface will prove to be viable for low-cost and high energy efficiency wireless communications and sensing systems.

## Methods

### $MoS_2$ ink and Graphene ink preparation

$MoS_2$ ink: This was achieved by liquid-phase exfoliation of bulk $MoS_2$ powder ($<2\,\mu m$, 99%) in water assisted by the use of pyrene−1-sulfonic acid sodium (PSI), >97%, purchased from Sigma Aldrich[20,33]. Bulk $MoS_2$ powder and PS1 were dispersed in de-ionized water at a concentration of 3 mg $mL^{-1}$ and 1 mg $mL^{-1}$, respectively. The dispersion was sonicated at 10 °C for 120 h. The resultant dispersion was centrifuged at 3500 rpm (1315 g) for 20 mins to remove the precipitated bulk material and then centrifuged at 15,000 rpm (20440 g) for 60 mins, twice, to remove excess PS1 from the dispersion. After washing, the precipitate was redispersed in the printing solvent. The solvent consisted of less than 1:10 propylene glycol:water by mass, ≥0.06 mg ml−1 Triton x−100 and ≥0.1 mg ml−1 xanthan gum[34]. The final ink concentration is diluted

to 2 mg $mL^{-1}$. The concentration of the resultant ink was assessed using a Varian Cary 5000 UV−vis spectrometer and by using the Beer−Lambert law and an extinction coefficients of 3400 L $g^{-1}$ $m^{-1}$ (at 672 nm) for $MoS_2$[20].

Graphene ink for screen printing: The graphene ink was made using a natural Dihydrolevoglucosenone (Cyrene) solution by Shear mixing[32,35]. Cyrene is a cellulose-derived solvent that is non-toxic, environmentally friendly, and sustainable. However, Cyrene solution is very expensive, so in this work the ink preparation was based on recycled Cyrene solution to save costs. 5 g expanded graphite flakes were obtained from Sigma-Aldrich (+50 mesh flake size). After rinsing and drying, the graphite flakes were added to 500 ml of Cyrene (Dihydrolevoglucosenone acid, >99%, Circa Group Pty Ltd.). The shear mixing procedure was then carried out using a specially designed shear mixer (L4R, Silverson) at 8000 rpm (6797 g), and the temperature was maintained at 10 °C using a cooling system with water circulation. After shear mixing, the 500 ml of mixture is centrifuged at 12,000 rpm (15294 g) and 10 °C for 1 h, and the tubes will be stratified. 400 ml of the top translucent pale yellow Cyrene solution was separated, which can be recycled and reused. At the same time, the bottom graphene ink was obtained. After 10 cycles of the reused process, 1000 ml of graphene ink was obtained, while only 1400 mL of Cyrene solution was used[32].

### Inkjet-printed zero-static power $MoS_2$ switch and screen-printed graphene metasurface patterns

Inkjet-printed $MoS_2$ switch: Paper (PEL P60, from Printed Electronics Ltd) was used as substrate[20]. Both the top and bottom electrodes were Ag and 4 printing passes were deposited. For the $MoS_2$ layer 80 printing passes were deposited. A Fujifilm Dimatix DMP 2800 model printer was used to carry out the printing of the 2D material heterostructures devices. The drop spacing is set at 20 μm for printing on paper. A step-by-step annealing procedure was employed[24]. The Ag ink was annealed at 150 °C for 30 mins under vacuum while for the $MoS_2$ ink, 90 mins are used.

Screen-printed graphene metasurface: The resolution of the screen printing dictates the mesh size required. Accordingly, the concentration of the graphene ink needs to be adjusted by adding Cyrene and properly mixing up with graphene nanoflakes. The 62-mesh screen was used. Before printing, the ink was mechanically agitated for 5 mins. A quick vacuum treatment was also applied to get rid of ink bubbles. To create a laminate that is equally distributed throughout, the capillary film (ULANO, EZ50-Orange) has been specifically chosen for this work. Three layers of the capillary film was fixed to the screen after a 120-mins exposure (40 mins per layer). The graphene ink was printed onto the A4 paper substrate (Xerox Performer A4 Paper 80 GSM) using a semi-automatic screen printer (YICAI-4060DV). The printed patterns were then dried and annealed in a vacuum furnace for 5 h at 100 °C. The printed patterns are then rolled by a rolling mill (Agile F130 Mill) to produce good electrical conductivity.

### Metasurface measurement

Two linearly polarized horn antennas (Aaronia AG, PowerLOG 70180) with a frequency range of 700 MHz–18 GHz were used as transmitter and receiver and connected to Keysight N9918A vector network analyzer (VNA). During the measurement, the time gating function of the VNA was used to remove any reflection and interference noise beyond the metasurface. The position of the transmitting antenna was fixed during the measurement and the receiving antenna was rotated from 0° – 180° to obtain the far field patterns.

### XPS measurement

A special designed $Au/MoS_2/Ag$ device was printed using Au probe directly as top electrode in order to remove the Au top electrode after

biasing, allowing us to directly detect the $MoS_2$ working interface. XPS spectra were acquired using Kratos Axis Ultra which equipped with monochromatic Al Ka source (hν = 1486.7 eV), under 8 E-9 mbar vacuum. Survey spectra and core-level spectra (O 1 s, C 1 s, Mo 3 d and S 2p region) were acquired using parameters of 80 eV pass energy, 0.5 eV step size and 20 eV pass energy, 0.1 eV step size, respectively. Neutralizer was applied during the experiment to eliminate the surface charging effect. Quantification of XPS spectra was performed using CasaXPS software. The relative sensitivity factors (RSFs) used for quantification are exported from casaXPS_KratosAxis-C1s which is a build-in library in CasaXPS. All spectra were calibrated to contaminated hydrocarbon components in C 1 s spectra located at 284.8 eV. Shirley backgrounds were applied to each spectrum for further peak fitting.

## Raman spectroscopy

A Renishaw Invia Raman spectrometer, equipped with a laser with excitation wavelength of 514.5 nm (with a power on the sample of 97.5 μW) and 633 nm (with a power on the sample of 44.8 μW), a 100X NA objective lens and 2400 grooves mm$^{-1}$ grating (at 514.5 nm) and 1800 grooves mm$^{-1}$ grating (at 633 nm), was used. The Raman spectra were collected on 10 different spots for each sample.

## Scanning electron microscopy

TEM sample is prepared using Helios nanolab 660 focused ion beam-scanning electron microscopy (FIB-SEM), with 2.5 nA milling current at 30 kV, 40 pA thinning current at 30 kV and 27 pA final cleaning current at 2 kV to minimize the FIB damage, redeposition and generation of amorphous layer to the sample. High-angle annular dark-field (HAADF) images, energy-dispersive X-ray (EDX) spectra and scanning pression electron diffraction (SPED) dataset are acquired using Talos F200X, which equipped with Super-X EDX detectors and Merlin diffraction camera. For STEM-EDX images, the probe current was measured to be 260 pA with probe size estimated to below 2.5 nm. The SPED dataset was acquired using gun lens 5, spot size 8, C2 aperture 10 μm with 2 nm step size, 0.29-degree pression angle, 5 pressions per step and 0.1 s dwell time as parameters, afterwards, processing using ACOM package. SEM-EDX characterisation were conducted using Magellan FEG-SEM (FEI) equipped with a silicon drift detector (SDD, X-Max 80 mm, Oxford Instrument) for EDX analysis. The accelerate voltage was adjusted between 10 – 15 kV to control the sampling depth. Half-device ($MoS_2$ on Ag) samples were coated with carbon to reduce the charging effect. The circuit was made using a gold probe on the surface of $MoS_2$ and an electrode connected to the Ag wire underneath.

## Data availability

Detailed experimental data are available in Supplementary Data1. More data are available from the corresponding author upon request.

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

## Acknowledgements

This work has been partially supported by UK Engineering Physical Sciences Research Council (EP/N010345 and EP/X028844/1) and ERC PEP2D (Contract No. 770047). Z.P. acknowledges financial support by the CSC. J.W. acknowledges financial support by the National Physical Lab of London. K.S.N. acknowledges the support from the Ministry of Education, Singapore (EDUNC-33–18-279-V12), the National Research Foundation, Singapore (AISG3-RP-2022-028) and the Royal Society UK (RSRP\R\190000). Authors would also like to thank Circa Group for providing Cyrene for the experiments.

## Author contributions

X.X. designed the reconfigurable metasurface, measured, analyzed the experimental data and draft the manuscript. Z.P. prepared $MoS_2$ ink, designed, printed the zero-static power $MoS_2$ switches and reviewed the draft. Z.Z. screen printed the metasurface. X.Z. prepared the graphene ink. Y.L. and X.L. conducted STEM measurement. J.W. did Raman. H.L. reviewed the draft. K.S.N. provided advice and reviewed the draft, C.C. supervised the project and reviewed the draft. Z.H. initiated and supervised the project, analysed the experimental data, and edited the manuscript. All authors have given approval to the final version of the manuscript.

## Competing interests

The authors declare no competing interest.
