## [Transparent Peer Review file · Nature Communications]

Fully printed zero-static power MoS₂ switch coded reconfigurable graphene metasurface for RF/microwave electromagnetic wave manipulation and control

Corresponding Author: Professor Zhirun Hu

Version 0:

Reviewer comments:

Reviewer #1

(Remarks to the Author)

The manuscript presents the concept of zero-static power switches, focusing on printable Ag/MoS₂/Ag vertical diodes as a solution to minimize power consumption. The incorporation of these diodes into a flexible metasurface, along with graphene patches, for manipulating electromagnetic waves is noteworthy. The practical significance of the research is emphasized by its potential applications in wireless communications, sensing, and holography.

However, before recommending acceptance, I request that the authors address the following concerns:

1. The authors emphasize that the average on-resistance of the printed Ag/MoS₂/Ag vertical diodes is about 10 Ω , which is the lowest on-state resistance among all printed inorganic memristive switches. However, such a low on-resistance can lead to a higher on-current at the set voltage (~ 1.75 V), resulting in elevated Joule heating during the device's activation process, potentially affecting the switch's cycle life. Is it possible that, due to this reason, the authors only reported 100 switching cycles in the manuscript?
2. The characterization provided by the authors regarding the transition of the 2H-phase MoS₂ to the 1T-MoS₂ phase, assisting in the formation of Ag-conductive filaments, lacks sufficient evidence and is not convincing. Could Ag ions diffuse between the stacking faults and defects in MoS₂ flakes to also form conductive filaments?
3. Please check Equation S(4).
4. The performance of the coded metasurface is limited.

(Remarks on code availability)

Reviewer #2

(Remarks to the Author)

This manuscript presents a fully printed zero-static power reconfigurable graphene metasurface on flexible substrate. The Ag/MoS₂/Ag structure is used as switch to active control the phase of reflected wave. A 3*3 metasurface is fabricated for 3.94 GHz, no power is required for keeping the state of the metasurface. Experiments are carried out to demonstrate the reconfigurable ability. The device can be used for wireless communication, sensing, and even holography. The work is interesting. However, there are still some points should be clarified:

1. Some detail information about the measurement should be given. How to apply the voltage on the Ag/MoS₂/Ag diode?
2. What is the switch speed of this device? What is expected life of the device?
3. For the metasurface cell, what is reflection amplitude? Since 3*3 cells are employed, what does the coding sequence '111' mean? Why do the coding sequences '000' and '111' generate different reflection angle?
4. Since the working wavelength is 76.1 mm, the size of the metasurface is 54.4*3 mm, which is only 2 times of the working wavelength, can this device work well? What is the efficiency of this device?

(Remarks on code availability)

Reviewer #3

(Remarks to the Author)

In this study, the authors prepared all-printed switches with MoS₂ nanosheets and silver. The switching mechanism was explored and discussed. The device was further integrated with printed graphene metasurfaces on flexible substrates, demonstrating their potential in high frequency applications.

Overall, this work shows such switches and metasurface can be made by all-printing-based methods. However, similar studies have been conducted in recent years and the novelty of this study is compromised. It is recommended that the manuscript should be submitted to specialized journals in the corresponding fields.

To improve the manuscript, the research progress on the nanosheet-based switches and on the printed graphene metasurface should be fully given to highlight the novelty of the current research. Further, extensive discussion on the obtained experimental results is required. The authors should also address the following questions.

1. The authors claimed that "such devices usually rely on new materials which are rarely directly compatible with the traditional CMOS technology". Why new materials are not compatible with the traditional CMOS technology? Refer to what kinds of new material.
2. The authors said that "printable electronics imposes much less strict towards the use of new materials". Not all new materials can be processed in solutions and the solution-processing (printing) requires tuning many parameters to realize good quality thin films.
3. In the abstract, the author named Ag/MoS₂/Ag as diodes. However, diodes refer to the device that can rectify current, i.e., only allowing current to flow in one direction.
4. In the abstract, RF is given without its full name.
5. What other materials were explored as zero-static power switches previously and what are the advantages of using 2D materials for this application?
6. Please give full name of IoT
7. Why choose 2.185 μm thickness for the MoS₂ layer? Will the device performance change when changing the MoS₂ layer thickness?
8. In 2014, Bessonov et al. fabricated very similar devices with a structure of Ag/MoS₂/Ag and observed the resistive switching behavior depending on the stoichiometry of the MoS₂/MoO_x (DOI: 10.1038/nmat4135). In 2019, Feng et al. fabricated Ag/MoS₂/Ag memristor and the resistance can be tuned between 10 to 1010 Ohm (DOI: 10.1002/aelm.201900740). A recent work by Saha et al. showed that MoS₂ based memristors exhibit superior performance with only a few picojoules energy cost (DOI: 10.1021/acsnano.3c10775). Will it be a similar mechanism as mentioned in the above literatures? What is the novelty for your devices compared with their work?
9. For the fabricated device (Ag/MoS₂/Ag), what is the reason that it can realize the zero-static power, while the others cannot? How was the device operated with a zero-static power in this study?
10. It is true that a single layer 2D material can prevent ions from diffusing. However, in the case of the nanosheet network, as observed in the cross-sectional SEM images, these nanosheets obviously form porous structures that ions can easily transport through these pores.
11. Since an array of switches are fabricated, why not show I-V curves of these devices to demonstrate the reproducibility?
12. In Fig. S7, are these two SEM images collected at the identical region of the MoS₂ layer before and after switching?
13. Please give scale bar in Fig. 5(b) and (c). Please use arrows to indicate each material in these figures.
14. The characters in Fig. 5(d) are not clear.
15. It is difficult to distinguish the far field pattern induced by the fabricated metasurface. What is the expected far field pattern with the current design of the metasurface?

(Remarks on code availability)

Version 1:

Reviewer comments:

Reviewer #1

(Remarks to the Author)

The response letter and the revisions made to the manuscript have addressed and satisfied my concerns. This manuscript can be accepted without further modifications.

Reviewer #2

(Remarks to the Author)

The authors have already revised the manuscript according to my comments, however, the replies for some questions are

not satisfied.

About the life time of the device, the authors do not give an evaluation. Figure 1 shows the changes in switch resistance after different cycles. It can be seen that the voltage of the resistance change is not stable. What is the reason for this? Is the process of resistance change related to the current carrying capacity of the sample? Is there an intermediate state related to voltage?

About the function of the metasurface, the coding sequence '111' and '000' should generate the same reflection angle because $d\phi(x)/dx$ are same. The authors do not give me a reasonable explanation. The size of the device is too small, the edge of device may cause the extensive diffraction. Therefore, I strongly recommend the authors to fabricate a device with large size (more cells) to get the demonstration again.

Reviewer #3

(Remarks to the Author)

The authors clearly stated the novelty of the research and addressed the comments. The manuscript meets the requirement for publication.

Version 2:

Reviewer comments:

Reviewer #2

(Remarks to the Author)

Since the authors have already added the experiment to demonstrate the validity of the device, this work can be accepted for publication.

Responses to the reviewers' comments

Manuscript ID: NCOMMS-23-59519-T

Title: Fully printed non-volatile MoS₂ coded reconfigurable graphene metasurface for RF/microwave electromagnetic wave manipulation and control

We very much appreciate reviewers' comments and suggestions. We have incorporated the suggestions and revised the manuscript accordingly. Those changes are highlighted in the revised manuscript. The following provides our point-by-point response to the reviewers' comments and concerns.

1 reference here is the number in the manual script.

1 reference here is the number in the supplementary information.

Reviewer #1:

The manuscript presents the concept of zero-static power switches, focusing on printable Ag/MoS₂/Ag vertical diodes as a solution to minimize power consumption. The incorporation of these diodes into a flexible metasurface, along with graphene patches, for manipulating electromagnetic waves is noteworthy. The practical significance of the research is emphasized by its potential applications in wireless communications, sensing, and holography.

However, before recommending acceptance, I request that the authors address the following concerns:

1. The authors emphasize that the average on-resistance of the printed Ag/MoS₂/Ag vertical diodes is about 10 Ω , which is the lowest on-state resistance among all printed inorganic memristive switches. However, such a low on-resistance can lead to a higher on-current at the set voltage (~ 1.75 V), resulting in elevated Joule heating during the device's activation process, potentially affecting the switch's cycle life. Is it possible that, due to this reason, the authors only reported 100 switching cycles in the manuscript?

Responses from the authors: We are grateful for this very important question. It's true that the lower the on-resistance, the higher the on-current. For memristive switches based on conductive filament formation, current limit must be set during the measurement (as well as in practical applications) to protect the devices from breaking down. We set 11 mA current limit, which is about the same comparing to those published works [Kim M, Ge R, Wu X, et al. 'Zero-static power radio-frequency switches based on MoS₂ atomrusters', *Nature communications*, 2018, 9 (1): 2524]. Under similar setting, the on-resistance of our printed devices is lowest. In the original manuscript, we have arbitrarily measured the devices up to 100 cycles manually. In the revised version, we have added extra measurements (**Fig. 1(b) and (c)**), whereas total of 300 cycles were tested manually. **Fig. 1(b) and (c)** now show the I-V characteristics and endurance of the device during these 300 cycles.

Fig. 1 (b) I-V characteristics measured over 300 cycles of an Ag/MoS₂/Ag memristive switch, (c) Endurance (resistance distribution) of the switch with 300 manual DC switching cycles.

To further clarify this, we have added the following sentence in the revised manuscript:

“**Fig. 1(b)** shows the I-V characteristics of an Ag/MoS₂/Ag memristive switch measured over 300 cycles, illustrating stable ON/OFF resistance and relatively small set voltage, comparable with values measured for most of the reported printed memristive switches (Table SI).”

“Overall, the printed MoS₂ switch shows performance comparable to the best printed switches reported in literature (Table SI, and Fig. S3); to note that previous devices were printed on silicon or plastic, while in our case the device is printed on paper.”

Table SI. State of the art of memristive switches fabricated by solution-based inorganic materials using printing methods.

Structure	Printing technique	LRS resistance	Set/Reset voltage	Cycle endurance	Ref
Ag/MoS ₂ /Ag	Inkjet printing	10 ¹ Ω	Set 1.75 V; Reset -0.7 V	300 cycles	Our work
Ag/ZnO/Ag	Electrohydrodynamic printing, spin coating	10 ⁶ Ω	Set 2 V; Reset -2 V	N/A	1
Ag/ZnO/Cu	Electrohydrodynamic Printing	10 ³ Ω	Set 1.25 V; Reset -1.25 V	500 cycles	2
Ag/TiO ₂ /Cu	Electrohydrodynamic Printing	10 ² Ω	Set 0.7 V; Reset -0.7 V	N/A	3
Ag/ZnSnO ₃ /Ag	Screen Printing, Electrohydrodynamic Atomization	10 ⁷ Ω	Set 2 V; Reset -2 V	100 cycles	4
Au/Cu-SiO ₂ NWs/Cu	Aerosol-Jet Printing	10 ⁴ Ω	Set 3 V; Reset -3 V	10 ⁴ cycles	5
Ag/a-TiO ₂ /Ag	Inkjet Printing	10 ⁷ Ω	Set 10 V; Reset -10 V	10 ³ cycles	6
Ag/ZnO/Ag	Electrohydrodynamic Printing	10 ² Ω	Set 3.75 V; Reset -3.75 V	10 ³ cycles	7
Ag/ZrO ₂ /Ag	Electrospray Deposition, Electrohydrodynamic Printing	10 ² Ω	Set 3.8 V; Reset -2.6 V	N/A	8
Ag/h-BN/Ag	Inkjet Printing	10 ⁵ Ω	Set 2 V; Reset -1 V	10 ³ cycles	9
Ag/MoS ₂ /Ag	Aerosol-Jet Printing	10 ¹ Ω	Set 0.18 V - 0.30 V, Reset -0.1 V	100 cycles	10
Ag/WSe ₂ /Ag	Aerosol-Jet Printing, Pneumatic atomizer	10 ⁵ Ω	Set 0.7 V; Reset -0.25 V	N/A	11
Ag/TiO ₂ /Carbon	Screen Printing, Inkjet Printing	10 ³ Ω	Set 1 V; Reset -3 V	100 cycles	12
Ag/ZrO ₂ /Ag	Electrohydrodynamic Printing	10 ⁵ Ω	Set 3 V; Reset -3 V	100 cycles	13
Ag/Cr-N-doped TiO ₂ /Ag	Reverse offset printing, EHD Printing, EHDA Printing	10 ⁴ Ω	Set 1 V; Reset -1 V	500 cycles	14
Ag/MoS ₂ /Gr	Inkjet printing	10 ⁴ Ω	Set 2 V; Reset -0.24 V	100 cycles	15

Fig. S3 Benchmarking of the on-state resistance of Ag/MoS₂/Ag switch compared to state-of-the-art printed inorganic memristive switches.

2. The characterization provided by the authors regarding the transition of the 2H-phase MoS₂ to the 1T-MoS₂ phase, assisting in the formation of Ag-conductive filaments, lacks sufficient evidence and is not convincing. Could Ag ions diffuse between the stacking faults and defects in MoS₂ flakes to also form conductive filaments?

Responses from the authors: We thank the Reviewer for the observation; there seemed a misunderstanding on the switching mechanism explanation, which we hope to clarify in our answer and in the revised text. The conductive filament formation is an accepted mechanisms for the switching behaviour, as already reported in the literature for printed MoS₂ memristive switches, e.g., [Feng, X. et al. ‘A fully printed flexible MoS₂ memristive artificial synapse with femtojoule switching energy’, *Adv. Electron. Mater.* 5, 1900740 (2019)], which experimentally demonstrates Ag ion diffusion in forming the conductive filaments. What we have proposed (based on our experiments) is that localised phase change (from 2H to 1T) could have assisted Ag diffusion and migration as some parts of MoS₂ become conductive due to phase change. We have performed more measurements on both pristine and used (switched on/off 20 times and at on-state, i.e., LRS) memristive switches.

We have revised the text to reflect these added experimental findings as following:

“To further investigate the mechanism of conductive filament formation, the changes in Ag content were measured by comparing a pristine sample to a used sample (switched on/off 20 times, the sample was at on-state, i.e., LRS). It is found that Ag content increases from 0 at. % to 0.3 at. % in the middle part of the MoS₂ layer, as depicted in **Fig. 3(c)**, indicating Ag diffusion and migration into the MoS₂ layer upon application of the external electric field. To note that the middle part of the MoS₂ layer was selected on purpose to avoid artefacts due to Ag that may have already migrated into MoS₂ layer from the Ag/MoS₂ interface during the inkjet printing process, i.e., before the bias is applied to the device. This result further confirms that the device switching mechanism is based on electro-migration^{31,32}.

However, the conventional conductive filament formation mechanism, which has been observed also in metal oxide memristors²⁰, might not work as well in printed layers of nanosheets: it is well known that 2D materials are difficult for ions to penetrate due to the high energy barrier caused in dense structure²². On the other hand, 2H MoS₂ is known to change its phase into the metallic 1T phase upon external stimuli such as ion intercalation and strain^{33,34}. Therefore, HAADF-STEM on the pristine and used sample, **Fig. 3(e) and (f)**, was conducted. Different lattice parameters for the MoS₂ layers can be observed, indicating the formation of localised 1T-MoS₂ phase. Although reactions between Ag and bulk MoS₂ is not thermodynamically favourable and no reactions can be observed at the interface by XPS measurements²³, the MoS₂ nanosheets show intrinsic defects including lattice bending, stacking faults and expansion of the MoS₂ layer structure after the voltage is applied (**Fig. S12 (a) and (b)**), which can generate strain in the MoS₂ flakes (**Fig. S8**), leading to formation of localised 1T-MoS₂ phase²⁴ and to the variance in the electric properties²⁵. Our measurements reveal that while a complete phase change in MoS₂ (from 2H to 1T) is highly unlikely, some localized phase change due to applied electric field as well as intrinsic defects of inkjet-printed MoS₂ may have catalysed the formation of Ag conductive paths to enable the device show macroscopic conductivity at on-state.”

We have also added the additional measurements in the **Supplementary Information** as following:

“Raman spectroscopy has been performed on both pristine and used memristive switches on the cross section, by cutting the device with a FIB. The spectrum taken on the switched memristive switch shows a slightly more pronounced peak at 286 cm⁻¹, **Fig. S14 (b)**, which may be attributed to the normally forbidden E_{1g} mode¹⁶. The relative intensity ratio of the peaks at 226 cm⁻¹ and at 406 cm⁻¹ is increasing after switching. This indicated that the MoS₂ flakes are fragmented after biasing, resulting in the reduction in the crystalline region size¹⁷ in agreement with the results presented in **Fig. S14 (e), (f) and Fig. S13.**”

Fig. S14 (a) Optical figure of an Ag/MoS₂/Ag memristive switch cross section. (b-c) Raman spectra of the MoS₂ cross section taken at 514.5 and 633 nm laser wavelength. (d) Normalized Raman spectra using the peak at 408 cm⁻¹. (e) HRTEM image of the pristine Ag/MoS₂/Ag cross-section. (f) HRTEM image of the Ag/MoS₂/Ag cross-section switched to LRS.

“To study Ag migration process through MoS₂ film, we have looked at changes of Ag 3d peak before and after switching the device. A special designed Au/MoS₂/Ag memristive switch which uses Au probe as one of the electrodes was printed. The structure allows us to directly investigate the working interface of MoS₂ after biasing. **Fig. S15** shows the XPS spectra for Ag 3d obtained from both pristine Au/MoS₂/Ag memristive switch and used one (at LRS). As it can be seen in **Fig. S15 (b)**, prominent Ag 3d doublets with binding energy values at 367.9 eV is shown, corresponding to Ag⁺ 3d_{5/2}¹⁸. This indicates that Ag appears at the interface of Au/MoS₂, which is clear evidence that Ag diffuses and migrates through the MoS₂ layer and reaches the top electrode. In addition, **Fig. S16** shows the SEM and EDX images of MoS₂/Ag layers after removing the Au probe and biasing. Ag elements can be directly detected from EDX mapping.”

Fig. S15 XPS spectra of the pristine and switched Au/MoS₂/Ag memristive switches' Ag 3d peaks.

Fig. S16 SEM and EDX image of the MoS₂/Ag device after biasing. (a) SEM image of the MoS₂/Ag layers after biasing and (b) Ag, Mo and S elements EDX mapping of the MoS₂/Ag device.

We've added the following text to the **Method** part of the manuscript:

“XPS measurement

A specially designed Au/MoS₂/Ag device was printed using Au probe directly as top electrode in order to remove the Au top electrode after biasing, allowing us to directly analyze the Au/MoS₂ interface. XPS spectra were acquired using Kratos Axis Ultra equipped with monochromatic Al Kα source (hν=1486.7 eV), under 8 E-9 mbar vacuum. Survey spectra and core-level spectra (O 1s, C 1s, Mo 3d and S 2p region) were acquired using parameters of 80 eV pass energy, 0.5 eV step size and 20 eV pass energy, 0.1 eV step size, respectively. Neutralizer was applied during the experiment to

eliminate the surface charging effect. Quantification of XPS spectra was performed using CasaXPS software. The relative sensitivity factors (RSFs) used for quantification are exported from casaXPS_KratosAxis-C1s which is a built-in library in CasaXPS. All spectra were calibrated to contaminated hydrocarbon components in C 1s spectra located at 284.8 eV. Shirley backgrounds were applied to each spectrum for further peak fitting.”

“Raman Spectroscopy

A Renishaw Invia Raman spectrometer, equipped with a laser with excitation wavelength of 514.5 nm 97.5 uw and 633 nm 44.8 uw laser power, a 100X NA, 0.85 objective lens and 2400 grooves mm⁻¹ grating (514.5 nm) and 1800 grooves mm⁻¹ grating (633 nm), were used. The Raman spectra were collected on 10 different spots for each sample and averaging to check 2H and 1T phase of MoS₂.”

“Scanning electron microscopy

SEM-EDX characterization was conducted using Magellan FEG-SEM (FEI) equipped with a silicon drift detector (SDD, X-Max 80 mm, Oxford Instrument) for EDX analysis. The accelerating voltage was adjusted between 10 to 15 kV to control the sampling depth. Half-device (MoS₂ on Ag) samples were coated with carbon to reduce the charging effect. The circuit was made using a gold probe on the surface of MoS₂ and an electrode connected to Ag underneath.”

3. Please check Equation S (4).

Responses from the authors: Thanks for having spotted this mistake. We’ve corrected the wrong.

$$[T_{Measurement-Dut}] = [T_{Microstrip\ line\ left}]^{-1} * [T_{Measurement}] * [T_{Microstrip\ line\ right}]^{-1} \quad S (4)$$

4. The performance of the coded metasurface is limited.

Response from the authors: With all due respect, we would argue that the coded metasurface prototype has performed as expected – providing reconfigurability to manipulate and control the reflected electromagnetic wave according to coding sequence, e. g., steering the beam to the desired directions (main beam reflected at about 40°, Fig. 5 (e), and at about 70°, Fig. 5 (f), for the same incident angle), or reducing the RCS by levelling the reflection in all directions (Fig. 5 (g) and (f)). While we are encouraged by the performance of the fully printed reconfigurable metasurface, we are aware that there is a room (a big room) for improvement in all aspects, such as ink preparation to further reduce the sheet resistance and costs, non-volatile switch design and fabrication to further reduce the on-resistance and increase FOM. We are enduring working on these improvements.

Referee #2:

This manuscript presents a fully printed zero-static power reconfigurable graphene metasurface on flexible substrate. The Ag/MoS₂/Ag structure is used as switch to active control the phase of reflected wave. A 3 x 3 metasurface is fabricated for 3.94 GHz, no power is required for keeping the state of the metasurface. Experiments are carried out to demonstrate the reconfigurable ability. The device can be used for wireless communication, sensing, and even holography. The work is interesting. However, there are still some points should be clarified:

1. Some detailed information about the measurement should be given. How to apply the voltage on the Ag/MoS₂/Ag diode?

Responses from the authors: We have added **Fig. S4** to illustrate how the DC characteristics were measured. The voltage was applied to the device through the probes which are connected to the DC supply.

We have added the additional measurement information in the **Supplementary Information** as following:

“FieldFox Vector Network Analyzer N9918A was used for measuring RF properties of Ag/MoS₂/Ag memristive switches, shown in **Fig. S5**. For the metasurface, the bias was provided by Agilent 33210A Function Waveform Generator through the bias lines on the metasurface. The sheet resistance of the graphene layer was measured using 4-point probe station (Jandel, RM3000).”

Fig. S4 Device DC characteristic measurement set up.

Fig. S5 RF measurement set up.

2. What is the switch speed of this device? What is expected of the life of the device?

Responses from the authors: We have measured the switch speed. The results are given in **Fig. S10** and **Fig. S11**.

Fig. S10 Switch speed measurement design.

Fig. S11 (a) Applied and responding pulse voltages of an Ag/MoS₂/Ag device (switch ON) and **(b)** switch OFF with different pulse voltages.

We have added the additional measurement information in the **Supplementary Information** as following:

“The schematic diagram of switch speed measurement setup is depicted in **Fig. S10**. The measurement setup involves monitoring the voltage across the resistor connected in series and the source voltage from the Agilent 33210A Function Generator and the IRF520 MOSFET switching circuit. For both turn-on and turn-off assessments, the series resistors are set at 100 Ω.

Fig. S11 (a) displays the turn-on waveform. The 100 Ω series resistor serves to limit the current and protect the system. The Ag/MoS₂/Ag device activates with a rise time of approximately 0.89 μs. **Fig. S11 (b)** depicts the output voltage changes of the switch over time following a turn-off pulse initiated at 10 μs. Each voltage trace exhibits a sharp negative spike at a specific moment, after which it remains steady for a period before instantly returning to zero, indicating the switch's closing moment. The turn-off time varies from 3.7 to 14.8 μs. Notably, the turn-off time depends on the magnitude of the applied turn-off pulse, lengthening as the pulse voltage decreases. Increasing the turn-off pulse voltage can decrease the turn-off time.”

Regarding the expected life of the device, please refer to **Fig.1(d)** in the manuscript. It shows that the retention of the device for both high resistance state (HRS) and low resistance state (LRS) is more than 10⁷ seconds, meaning the device can work stably in near 7 months. The device is still working at the time we write this response. The reliability of 2D materials devices is a very challenging issue. There is no theory that can be used to predict such devices' lifetime so far.

3. For the metasurface cell, what is reflection amplitude? Since 3 * 3 cells are employed, what does the coding sequence ‘111’ mean? Why do the coding sequences ‘000’ and ‘111’ generate different reflection angle?

Responses from the authors: We have added reflection amplitude in **Fig. S19**. At operating frequency point, the reflection amplitudes are almost the same for both State 0 and State 1 but with phase difference (between incident and reflected waves) of 180° (**Fig. 5(a)**).

Fig. 5 (a) Reflection phase of the unit cell. Fig. S19 (b) Reflection magnitude of unit cell.

When the non-volatile MoS_2 RF memristive switch works in different states, i.e., on or off, the reflection coefficients of the metasurface behave differently. Coding '1' and coding '0' refer to the on-state and off-state, respectively. Coding sequence '111' means that all switches in columns 1 (most left in **Fig. 5(b)**), 2 and 3 are switched on, whereas '000' means switches in these columns are all off. '010' means that all switches in column 1 are off, but on in column 2 and off in column 3. Different coding sequence combination can provide different reflection angles, as the reflection angles must obey generalized law of reflection and refraction [Yu N, Genevet P, Kats M A, et al. Light propagation with phase discontinuities: generalized laws of reflection and refraction. *Science*, 2011, 334(6054): 333-337.], explained as follows,

Fig. R1. Schematics used to derive the general Snell's law of refraction and reflection.

$$k_0 n_i \sin(\theta_i) dx + (\Phi + d\Phi) - (k_0 n_t \sin(\theta_t) dx + \Phi) = 0 \quad (1)$$

$$n_t \sin(\theta_t) - n_i \sin(\theta_i) = \frac{\lambda_0}{2\pi} \frac{d\phi(x)}{dx} \quad (2)$$

$$n_i \sin(\theta_r) - n_i \sin(\theta_i) = \frac{\lambda_0}{2\pi} \frac{d\phi(x)}{dx} \quad (3)$$

By introducing a sudden phase change, known as a phase discontinuity, at the boundary between two different materials, we can re-examine the principles of reflection and refraction through the lens of Fermat's principle. Let's consider a plane wave hitting the interface at an angle θ_i . Assuming that two paths are infinitesimally near the real path of light (refer to Fig. R1 and Equation 1), the phase difference between these paths is negligible. Where θ_t represents the refraction angle, Φ and $\Phi + d\Phi$ denote the phase discontinuities at the points where the two paths intersect the interface, and dx is the gap between these crossing points. The refractive indices of the two media are represented by n_i and n_t , where k_0 equals $\frac{\lambda_0}{2\pi}$, with λ_0 being the wavelength in vacuum. If the phase gradient along the boundary is kept constant, it results in the derivation of the generalized Snell's law of refraction. According to Equation 2 and 3, the direction of the refracted beam and reflect beam can be controlled to any desired angle by implementing a suitable, constant phase discontinuity gradient $\frac{d\phi(x)}{dx}$ along the interface. The existence of a non-zero phase gradient in this modified version of Snell's law means that different incidence angles, denoted as θ_i , lead to varying refraction and reflect angles.

4. Since the working wavelength is 76.1 mm, the size of the metasurface is 54.4 * 3 mm, which is only 2 times of the working wavelength, can this device work well? What is the efficiency of this device?

Responses from the authors: A metasurface is composed of many unit cells that are normally smaller than the wavelength of electromagnetic waves. In this work, the largest size of the unit cell is of 54 mm, which is smaller than the wavelength. Due to the limitation of our fabrication capability, we could have only printed 3 x 3 arrays. The Reviewer is right, the metasurface would perform better if there are more cells, i.e., large arrays [Wu, G.-B., Dai, J. Y., Cheng, Q., Cui, T. J. & Chan, C. H. Sideband-free space-time-coding metasurface antennas. *Nat. Electron.* **5**, 808–819 (2022).][Liang, J. C. *et al.* An Angle-Insensitive 3-Bit Reconfigurable Intelligent Surface. *IEEE Trans. Antennas Propag.* **70**, 8798–8808 (2022).]. By changing the cell sizes, shapes, materials and arrangement of these cells, the incident electromagnetic waves can be manipulated. Regarding the efficiency of the metasurface, this must be judged based on its functionality. For example, if the metasurface is mainly used for reflecting an incident wave to a desired direction, its efficiency is mainly determined by the reflectivity. If the main function is to absorb incident waves, RCS would be of interest. Our design mainly focuses on manipulating reflecting electromagnetic waves. At the operating frequency of 3.49 GHz, the reflection magnitude at the on-state is -2 dB, i.e., approximately 79.4% of the incident power is reflected. At the off state, the reflection magnitude is -0.9 dB, which means the efficiency is 90.2 %, meaning that 81.3 percent of the incident power is reflected.

Referee #3:

In this study, the authors prepared all-printed switches with MoS₂ nanosheets and silver. The switching mechanism was explored and discussed. The device was further integrated with printed graphene metasurfaces on flexible substrates, demonstrating their potential in high frequency applications.

Overall, this work shows such switches and metasurface can be made by all-printing-based methods. However, similar studies have been conducted in recent years and the novelty of this study is compromised. It is recommended that the manuscript should be submitted to specialized journals in the corresponding fields.

To improve the manuscript, the research progress on the nanosheet-based switches and on the printed graphene metasurface should be fully given to highlight the novelty of the current research. Further, extensive discussion on the obtained experimental results is required. The authors should also address the following questions.

Responses from the authors: With all due respect, we must disagree with the Reviewer on the novelty of this work. It is the first time that **(1) printed MoS₂ memristive switch (also known as memristor) has been measured, characterised and modelled at radiofrequency (RF) spectrum, (2) fully printed RF functional components on flexible substrate, where the passive elements, i.e., graphene patches, are screen-printed and the active elements (MoS₂ switches) are inkjet-printed on top of the passive elements, paving the way for low-cost, fully printed flexible electronics, and (3) fully printed non-volatile reconfigurable metasurface for wireless communications and sensing applications.** The integration of non-volatile switches (memristive switches are non-volatile) with metasurface is of significance in realizing net zero in wireless communication and sensing systems. In 6G, large intelligent reflective metasurfaces (IRS) will be deployed. Each IRS could have hundreds and thousands of elements, and each element could have a few switches for reconfigurability. A huge static power will be dissipated if current semiconductor switches (diodes or transistors) are to be used. Furthermore, thermal management for such large IRS will be challenging and costing. On the other hand, non-volatile switches proposed here consume no static power hence not only significantly increase the system power efficiency but also reduce the system complexity as thermal management can be significantly reduced (if still needed at all). In this work, we also propose, based on our measurement, that the phase change (from 2H to 1T) could have assisted Ag migration as some parts of MoS₂ become conductive.

Following the advice of the Reviewer, we have revised the manuscript to highlight the novelty of our research, as following:

“In contrast to previous works based on the use of commercially available diodes, made with traditional (rigid) semiconductors, we report a fully printed MoS₂ non-volatile switch enabled zero-static power reconfigurable graphene metasurface on a flexible substrate that is suitable for low-cost and disposable IoT applications.”

1. The authors claimed that “such devices usually rely on new materials which are rarely directly compatible with the traditional CMOS technology”. Why new materials are not compatible with the traditional CMOS technology? Refer to what kinds of new material.

Responses from the authors: A good example will be the integration of CVD graphene into CMOS technology. CVD graphene is compatible with large scale electronics (but both its thermal and electrical properties are not as good as exfoliated graphene), however there are several challenges when the material is integrated into CMOS processes: integration into a silicon chip requires to finely control the material growth and to minimize wrinkle formation and residue associated with the transfer process, that will affect the reproducibility (a review on the challenges of CVD graphene into CMOS technology can be found here: Akinwande, D., Huyghebaert, C., Wang, CH. et al. Graphene and two-dimensional materials for silicon technology. *Nature* 573, 507–518 (2019). <https://doi.org/10.1038/s41586-019-1573-9>). Why new 2D materials are not normally compatible with the traditional CMOS technology is due to (1) materials properties, (2) material preparations (such as material growth and transfer), and (3) fabrication and manufacturing processes.

(1) Material Properties:

Electronic properties of 2D materials can be very different to that of silicon. For instance, Graphene has zero bandgap but very high mobility. However, its mobility can be severely degraded when it is transferred to a silicon substrate. The other example is 2D black Phosphorus (BP) which is very sensitive to oxygen and easily oxidized at relatively low temperature. These materials' properties are not compatible to current CMOS technology.

(2) Material growth/preparations:

Currently most 2D materials, such as graphene and MoS₂, can only be obtained either through mechanical exfoliation or CVD growth. Mechanically exfoliated 2D materials can have very high quality (e.g., extremely high mobility for graphene) but not suitable for mass production. The latter can't provide the same quality as those of exfoliation (e.g., much low mobility). Even with today's technology, these 2D materials can't grow easily, directly on Silicon substrates. They need to be transferred to the host substrates which is not compatible with today CMOS manufacturing workflows.

(3) Fabrications and manufacturing processes:

(i) Patterning and Etching:

2D materials, such as graphene and MoS₂, react differently to chemical etchants compared to silicon, and due to their fragile physical properties, current CMOS lithography and etching processes will need adjustments to prevent material damage.

(ii) Interface issues:

When attempting to integrate 2D materials, such as graphene, with silicon, heterostructures are likely to be formed, which may cause issues such as high interface state density and charge trapping at the interface and require interface engineering to ensure device performance. This is again not compatible with current CMOS technology.

(iii) Packaging and interconnect technologies:

2D materials, such as, BP requires special packaging techniques to protect the material from damage due to its sensitivity to the environment, which is not compatible with current CMOS technologies.

In addition, 2D materials' scalability can also be challenging when they are implemented with current CMOS technologies.

To avoid misunderstanding, we revised the sentence as following:

“Still, the integration of such switches in real life electronics is slow because of challenges in compatibility of new materials with traditional CMOS technology. At the same time, printable electronics allows to go beyond the limits of CMOS technology by enabling low-cost processes at ambient temperature and integration of devices onto flexible substrates, such as paper and plastic.”

2. The authors said that “printable electronics imposes much less strict towards the use of new materials”. Not all new materials can be processed in solutions and the solution-processing (printing) requires tuning many parameters to realize good quality thin films.

Responses from the authors: We agree with the Reviewer that not all new materials can be processed into inks, however printing technologies enable cost reduction, compatibility with many substrates, including flexible ones, such as plastics and papers. A full library of metallic, semiconducting and insulating solution-processed materials is already available. Printed technology enables rapid prototyping of devices even at ambient conditions, low cost, and high speed on flexible and stretchable substrates, which is very attractive for wearable electronics and IoTs.

3. In the abstract, the author named Ag/MoS₂/Ag as diodes. However, diodes refer to the device that can rectify current, i.e., only allowing current to flow in one direction.

Responses from the authors: We are grateful to the referee for pointing this out. It would have been better named as memristive switches, providing low resistance (ideally 0 Ω) when it's on and very high resistance (ideally ∞ Ω) when off. On the other hand, the Ag/MoS₂/Ag heterostructure is equivalent to a double diode, as also observed for the same heterostructure made with graphene in place of silver [Fig. 4d, McManus, D. et al. Water-based and biocompatible 2D crystal inks for all-inkjet-printed heterostructures. *Nat. Nanotech.* 12, 343-350 (2017)].

4. In the abstract, RF is given without its full name.

Responses from the authors: The full name is radiofrequency (RF). Thanks for pointing it out. We have added ‘radiofrequency’ accordingly.

5. What other materials were explored as zero-static power switches previously and what are the advantages of using 2D materials for this application?

Responses from the authors: Zero-static power switches (also known as memristive switches or memristors) have been made using other materials [Pi S, Ghadiri-Sadrabadi M, Bardin J C, et al. Nanoscale memristive radiofrequency switches, *Nat. Commun.* 6:7519, doi: 10.1038/ncomms8519 (2015)] [Onofrio N, Guzman D, Strachan A. Atomic origin of ultrafast resistance switching in nanoscale electrometallization cells, *Nature Materials*, 2015, 14(4): 440-446]. They are, however, not printable. 2D material memristors based on monolayer or few layer MoS₂, graphene or other 2D materials have also been reported [Kim M, Pallecchi E, Ge R, et al. Analogue switches made from boron nitride monolayers for application in 5G and terahertz communication systems, *Nature Electronics*, 2020, 3(8): 479-485] [Kim M, Ge R, Wu X, et al. Zero-static power radio-frequency switches based on MoS₂ atomristors, *Nat. Commun.* 2018, 9(1): 2524.] [Wang M, Cai S, Pan C, et al. Robust memristors based on layered two-dimensional materials, *Nature Electronics*, 2018, 2(1): 130-136]. Yet again they are not printable. In our work, MoS₂ ink was used for the active switches and

graphene ink for the metasurface patches. These inks are biocompatible and are not based on toxic chemicals. The use of solution-processed 2D materials inks enable simple and low-cost fabrication techniques, such as inkjet and screen printing, and integration on flexible substrates, such as paper. As for this application, **Supplementary Information Table SIII** lists the recently published works on reconfigurable metasurfaces. They all use commercially available diodes, made with traditional (rigid) semiconductors. The distinguished feature of the coded metasurface reported in our work is that they consume no static power, significantly increasing the system energy efficiency. Furthermore, the whole reconfigurable metasurface has been fully printed on a flexible substrate, easily placed on the locations where the wireless channel needs to be enhanced or reconfigured.

Supplementary Table SIII. Static power consumption comparison

Ref	Number of Switch	Fabrication process	Static power supply (assuming half working)	Bias	Switch type	Reconfigurability	Material
19	400	PCB technology and layer stacking	5.8 W	5 V	MADP-000907-14020x	Yes	Copper
20	128	PCB technology and layer stacking	0.928 W	1.4 V	MADP-000907-14020W	Yes	Copper
21	160	PCB technology and layer stacking	4 W	0.8V	BAR50-02V	Yes	Copper
22	512	PCB technology and layer stacking	3.7 W	5 V	MADP-000907-14020x	Yes	Copper
23	18	PCB technology and layer stacking	1.08 W	5 V	BAR63-03W	Yes	Copper
24	1152	PCB technology and layer stacking	69.12 W	5 V	MADP-000907-14020x	Yes	Copper
Our work	9	Fully printed	0 W	1.7 V	Non-volatile MoS ₂ switch	Yes	Graphene

6. Please give full name of IoT

Responses from the authors: Internet of Things (IoT). We have added in the main manuscript accordingly.

7. Why choose 2.185 μm thickness for the MoS₂ layer? Will the device performance change when changing the MoS₂ layer thickness?

Responses from the authors: We have systematically studied the impact of MoS₂ layer thickness on the device performance by printing at the same concentration but changing the number of printed passes (PPs). We found that above 80 PPs, the device shows no shortcuts between the top and bottom Ag electrodes and has higher cycle endurance (**Fig. S2 (d)**) while having the lower set/reset voltages (average set voltage 2 V, reset voltage -1 V). Hence, we made all devices with 80 PPs, corresponding to a thickness of $\sim 2 \mu\text{m}$.

To clarify this, we have added **Fig. S2** in the **Supplementary Information** and the following sentence in the manuscript:

Fig. S2. Memristive switching characteristics of Ag/MoS₂/Ag switches, tested over 3 cycles, made with 2 mg/mL MoS₂ ink and printed in different PPs: (a) 20 PPs; (b) 40 PPs; (c) 60 PPs; (d) 80 PPs; (e) 90 PPs; (f) 100 PPs; (g) 120 PPs; (h) 140 PPs.

To avoid misunderstanding, we revised the sentence as following:

“The thickness of the MoS₂ layer is 2.185 μm. Note that thinner films have shown shortcuts between the top and bottom Ag electrodes, as shown in Fig. S2, hence this thickness was selected for all devices to gain as low set/reset voltages as possible while avoiding shortcuts.”

8. In 2014, Bessonov et al. fabricated very similar devices with a structure of Ag/MoS₂/Ag and observed the memristive switching behavior depending on the stoichiometry of the MoS₂/MoO_x (DOI: 10.1038/nmat4135). In 2019, Feng et al. fabricated Ag/MoS₂/Ag memristor and the resistance can be tuned between 10 to 1010 Ohm (DOI: 10.1002/aelm.201900740). A recent work by Saha et al. showed that MoS₂ based memristors exhibit superior performance with only a few picojoules energy cost (DOI: 10.1021/acsnano.3c10775). Will it be a similar mechanism as mentioned in the above literature? What is the novelty for your devices compared with their work?

Responses from the authors: Thank you for providing these references.

- In DOI: 10.1038/nmat4135, ‘A MoS₂ film is deposited by a spreading technique’, i.e., the device was not inkjet-printed. ‘Deposited by a spreading technique’ and inkjet-printing are two very different techniques.
- DOI: 10.1002/aelm.201900740 has indeed reported aerosol-jet printed MoS₂ volatile and non-volatile memristive switches. We have referred to this work (listed as [10]) in our original **Supplementary Information**. We are not claiming the novelty of our inkjet-printed switches in this report. The novelties of our works are (1) **it’s the first time that printed MoS₂ memristive switch (also known as memristor) has been measured, characterised and modelled at radiofrequency (RF) spectrum (so to significantly extend the applications of such printed devices),** (2) **it’s the first time that fully printed RF functional system on flexible substrate, where the passive elements, i.e., graphene patches, are screen-printed and the active, i.e., MoS₂ memristive switches are inkjet-printed on top of the passive elements, is reported, paving the way for low-cost, fully printed flexible electronics, and**

(3) it's the first time that fully printed non-volatile reconfigurable metasurface for wireless communications and sensing applications is presented.

- DOI: 10.1021/acsnano.3c10775 'demonstrate a high-performance multifunctional memristor using a thin film of liquid-phase exfoliated (LPE) 2D MoS₂ pinched between two electrodes.' The memristor reported in this referred work was not prepared using any printing techniques. This paper was accepted in December 19 and published December 21 2023. We didn't have a chance to refer this work as our original manuscript was submitted before this paper was published.

We have added DOI: 10.1038/nmat4135 and DOI: 10.1021/acsnano.3c10775 in our revised manuscript (ref. 15, 16).

9. For the fabricated device (Ag/MoS₂/Ag), what is the reason that it can realize the zero-static power, while the others cannot? How was the device operated with a zero-static power in this study?

Responses from the authors: The reason that it can realize the zero-static power is because that conductive filaments will form once the device is on. There is no need of any DC bias to sustain the on-state as conductive filaments will stay until the device is reversely biased, meaning there is no static power consumption hence zero-static power. This is very different comparing to, say, a PIN diode switch, where the diode must be biased with DC voltage to sustain its on-state hence dissipate static power. We would like to clarify that we are not suggesting that zero-static power consumption is exclusive to our device; indeed, other printed memristive switches (or more commonly known as memristors) listed in **Supplementary Information Table S1** can all realize the zero-static power due to the very same mechanism hence they are known as non-volatile switches. In general, semiconductor switches used in electronic circuits dissipate both dynamic power (to switch on or off) and static power (to sustain its on- or off-states). A non-volatile switch consumes dynamic power but need no power to sustain its on- or off-states hence zero static power. Comparisons of static power consumptions between this work and other published works on reconfigurable metasurfaces are provided in **Supplementary Information Table SIII**.

10. It is true that a single layer of 2D material can prevent ions from diffusing. However, in the case of the nanosheet network, as observed in the cross-sectional SEM images, these nanosheets obviously form porous structures that ions can easily transport through these pores.

Responses from the authors: We fully agree with the Reviewer that a network of nanosheets is characterized by a degree of porosity, hence enabling diffusion of ions. However, the film is very thick as compared to metal oxide memristor (less than a hundred nanometer), so the formation of the conductive filaments can be more difficult. This is why we have investigated if there exists an additional mechanism that could have assisted ion diffusion/migration through the layers. Please note that the phase change does not rule out conductive filament formation. Quite opposite, the phase change could have catalyzed the ion diffusion/migration (phase change observed in localized areas where filament formation is more likely to happen).

We have revised the text for better clarification as following:

“To further investigate the mechanism of conductive filament formation, the changes in Ag content were measured by comparing a pristine sample to a used sample (switched on/off 20 times, the sample was at on-state, i.e., LRS). It is found that Ag content increases from 0 at. % to 0.3 at. % in the middle part of the MoS₂ layer, as depicted in **Fig. 3(c)**, indicating Ag diffusion and migration into the MoS₂ layer upon application of the external electric field. To note that the middle part of the MoS₂ layer was selected on purpose to avoid artefacts due to Ag that may have already migrated into MoS₂ layer from the Ag/MoS₂ interface during the inkjet printing process, i.e., before the bias is applied to the device. This result further confirms that the device switching mechanism is based on electro-migration^{31,32}.

However, the conventional conductive filament formation mechanism, which has been observed also in metal oxide memristors²⁰, might not work as well in printed layers of nanosheets: it is well known that 2D materials are difficult for ions to penetrate due to the high energy barrier caused in dense structure²². On the other hand, 2H MoS₂ is known to change its phase into the metallic 1T phase upon external stimuli such as ion intercalation and strain^{33,34}. Therefore, HAADF-STEM on the pristine and used sample, **Fig. 3(e) and (f)**, was conducted. Different lattice parameters for the MoS₂ layers can be observed, indicating the formation of localised 1T-MoS₂ phase. Although reactions between Ag and bulk MoS₂ is not thermodynamically favourable and no reactions can be observed at the interface by XPS measurements²³, the MoS₂ nanosheets show intrinsic defects including lattice bending, stacking faults and expansion of the MoS₂ layer structure after the voltage is applied (**Fig. S12 (a) and (b)**), which can generate strain in the MoS₂ flakes (**Fig. S8**), leading to formation of localised 1T-MoS₂ phase²⁴ and to the variance in the electric properties²⁵. Our measurements reveal that while a complete phase change in MoS₂ (from 2H to 1T) is highly unlikely, some localized phase change due to applied electric field as well as intrinsic defects of inkjet-printed MoS₂ may have catalysed the formation of Ag conductive paths to enable the device show macroscopic conductivity at on-state.”

11. Since an array of switches are fabricated, why not show I-V curves of these devices to demonstrate the reproducibility?

Responses from the authors: The I-V curves of the device has already been presented in **Fig. 1** in the manuscript. We have added I-V curves for each individual device used in the array in **Supplementary Information Fig. S21**.

Fig. S21 I-V curve of arrays of switch.

12. In Fig. S7, are these two SEM images collected at the identical region of the MoS₂ layer before and after switching?

Responses from the authors: The regions are about the same but not identical as two devices were used for the measurement. For STEM measurement, the device must be cut to get the cross section. Once the unused device is cut, it can't work anymore. Both images were taken at very middle part of the devices.

To demonstrate that our analysis is solid, we have tested two more devices and added the results in **Supplementary Information Fig. S17** with a brief comment.

Fig. S17 TEM image of Ag/MoS₂/Ag devices. (a) unused (pristine) switch and (b) used switch (20 times on/off cycles). After applying bias on the device, the memristive switch shows a less ordered arrangement of MoS₂ flakes in the film and a looser structure, as compared to the pristine film.

13. Please give scale bar in Fig. 5(b) and (c). Please use arrows to indicate each material in these figures.

Responses from the authors: Thank you for your suggestions. We've acted accordingly.

Fig. 5 (b) Photos of the whole fully printed 3 × 3 zero-static power MoS₂ switch coded RF/microwave reconfigurable graphene metasurface and (c) its unit cell.

14. The characters in Fig. 5(d) are not clear.

Responses from the authors: Thanks. Updated accordingly.

Fig. 5 (d) Photo of the setup in the anechoic chamber

15. It is difficult to distinguish the far field pattern induced by the fabricated metasurface. What is the expected far field pattern with the current design of the metasurface?

Responses from the authors: We should have explained these better in the original manuscript. These are the expected results according to our design and simulation. The far field patterns can often be evaluated by their peak (main lobe) and side lobe values. The purpose of the coded metasurface is to provide reconfigurability to manipulate and control the reflected electromagnetic wave according to coding sequence. The metasurface can be used to steer the reflected beam to the desired directions (main beam reflected at about 40° , **Fig. 5 (e)**, and at about 70° , **Fig. 5(f)** for the same incident angle), or to reduce the RCS by levelling the reflection in different directions (**Fig. 5 (g) and (h)**).

Reference (supplement information)

1. Choi, K. H., Mustafa, M., Rahman, K., Jeong, B. K. & Doh, Y. H. Cost-effective fabrication of memristive devices with ZnO thin film using printed electronics technologies. *Appl. Phys. A* **106**, 165–170 (2012).
2. Muhammad, N. M. *et al.* Fabrication of printed memory device having zinc-oxide active nano-layer and investigation of resistive switching. *Curr. Appl. Phys.* **13**, 90–96 (2013).
3. Duraisamy, N., Muhammad, N. M., Kim, H.-C., Jo, J.-D. & Choi, K.-H. Fabrication of TiO₂ thin film memristor device using electrohydrodynamic inkjet printing. *Thin Solid Films* **520**, 5070–5074 (2012).
4. Siddiqui, G., Ali, J., Doh, Y.-H. & Choi, K. H. Fabrication of zinc stannate based all-printed resistive switching device. *Mater. Lett.* **166**, 311–316 (2016).
5. Catenacci, M. J. *et al.* Fully printed memristors from Cu–SiO₂ core–shell nanowire composites. *J. Electron. Mater.* **46**, 4596–4603 (2017).
6. Salonikidou, B. *et al.* Inkjet-printed Ag/a-TiO₂/Ag neuromorphic nanodevice based on functionalized ink. *Adv. Eng. Mater.* **24**, 2200439 (2022).
7. Rafique, A. F., Haji Zaini, J., Bin Esa, M. Z. & Nauman, M. M. Printed memory devices using electrohydrodynamic deposition technique. *Appl. Phys. A* **126**, 134 (2020).
8. Nauman, M. M., Zulfikre Esa, M., Zaini, J. H., Iqbal, A. & Bakar, S. A. Zirconium oxide based memristors fabrication via electrohydrodynamic printing. In *2020 IEEE 11th International Conference on Mechanical and Intelligent Manufacturing Technologies (ICMIMT)* 167–171 (IEEE, 2020).
9. Zhu, K. *et al.* Inkjet-printed h-BN memristors for hardware security. *Nanoscale* **15**, 9985–9992 (2023).
10. Feng, X. *et al.* A fully printed flexible MoS₂ memristive artificial synapse with femtojoule switching energy. *Adv. Electron. Mater.* **5**, 1900740 (2019).
11. Li, Y. *et al.* Aerosol jet printed WSe₂ crossbar architecture device on kapton with dual functionality as resistive memory and photosensor for flexible system integration. *IEEE Sens. J.* **20**, 4653–4659 (2020)
12. Lien, D.-H. *et al.* All-printed paper memory. *ACS Nano* **8**, 7613–7619 (2014).
13. Awais, M. N., Kim, H. C., Doh, Y. H. & Choi, K. H. ZrO₂ flexible printed resistive (memristive) switch through electrohydrodynamic printing process. *Thin Solid Films* **536**, 308–312 (2013)
14. Khan, M. *et al.* All-printed flexible memristor with metal–non-metal-doped TiO₂ nanoparticle thin films. *Nanomaterials* **12**, 2289 (2022).
15. Peng, Zixing, *et al.* "Fully printed memristors made with MoS₂ and graphene water-based inks." *Materials Horizons* (2024).
16. Frey, Gitti L., *et al.* "Raman and resonance Raman investigation of MoS₂ nanoparticles." *Physical Review B* 60.4 (1999): 2883.
17. Blanco, Élodie, *et al.* "Resonance Raman spectroscopy as a probe of the crystallite size of MoS₂ nanoparticles." *Comptes Rendus. Chimie* 19.10 (2016): 1310–1314.
18. Gaarenstroom, S. W., and N. J. T. J. Winograd. "Initial and final state effects in the ESCA spectra of cadmium and silver oxides." *The Journal of chemical physics* 67.8 (1977): 3500–3506.

19. Gros, J.-B., Popov, V., Odit, M. A., Lenets, V. & Lerosey, G. A reconfigurable intelligent surface at mmWave based on a binary phase tunable metasurface. *IEEE Open J. Commun. Soc.* **2**, 1055–1064 (2021).
20. Wang, Z. X. *et al.* A low-cost and low-profile electronically programmable bit array antenna for two-dimensional wide-angle beam steering. *IEEE Trans. Antennas Propag.* **71**, 342–352 (2023).
21. Trichopoulos, G. C. *et al.* Design and evaluation of reconfigurable intelligent surfaces in real-world environment. *IEEE open j. Commun. Soc.* **3**, 462–474 (2022).
22. Bai, X. *et al.* Radiation-type programmable metasurface for direct manipulation of electromagnetic emission. *Laser Photonics Rev.* **16**, 2200140 (2022).
23. Chaimool, S., Hongnara, T., Rakluea, C., Akkarackthalin, P. & Zhao, Y. Design of a PIN diode-based reconfigurable metasurface antenna for beam switching applications. *Int. J. Antennas Propag.* **2019**, 7216324 (2019).
24. Wang, C. *et al.* Reconfigurable transmissive metasurface synergizing dynamic and geometric phase for versatile polarization and wavefront manipulations. *Mater. Des.* **225**, 111445 (2023).

Responses to Reviewer#2's comments and suggestions

Manuscript ID: NCOMMS-23-59519-T

Title: Fully printed non-volatile MoS₂ coded reconfigurable graphene metasurface for RF/microwave electromagnetic wave manipulation and control

We would like to thank Reviewer#2's comments and suggestions. We have incorporated the Reviewer's suggestions and revised the manuscript accordingly. Any changes have been highlighted in the revised manuscript. The following provides our point-to-point response to the Reviewer's comments and suggestions.

Reviewer#2:

The authors have already revised the manuscript according to my comments, however, the replies for some questions are not satisfied.

Response from the authors: We thank the Reviewer for supporting our work. We have revised the manuscript following the Reviewer's suggestions.

About the lifetime of the device, the authors do not give an evaluation. Figure 1 shows the changes in switch resistance after different cycles. It can be seen that the voltage of the resistance change is not stable. What is the reason for this? Is the process of resistance change related to the current carrying capacity of the sample? Is there an intermediate state related to voltage?

We'll answer these questions one by one.

Q1: About the lifetime of the device, the authors do not give an evaluation.

Response from the authors: We have added lifetime of our device (**Fig. 1(d)**) and provided comparison of lifetimes between our device and other published works (**Table SI**). It can be observed that our device has been tested for 10⁷ s, the longest testing duration among the reported printed memristive switches or memristors. The requirement for data retention in commercialization is normally about ten years at ambient temperature [Sun W, et al. "Understanding memristive switching via in situ characterization and device modeling." *Nature Communications* 10.1 (2019): 3453.] and [Catenacci J, et al. "Fully Printed Memristors from Cu - SiO₂ Core - Shell Nanowire Composites." *Journal of Electronic Materials* 46 (2017): 4596-4603.]. By exponentially extrapolating the data out to 10 years (a method that has been widely employed to estimate retention times [Peng Z, et al. "Fully printed memristors made with MoS₂ and graphene water-based inks." *Materials Horizons*, 2024, 11(5): 1344-1353.]. We show that the switching on/off ratio of the device will remain average of 6×10^5 for over 10 years.

Fig. 1(d). Retention of the switch, measured over 10^7 s at room temperature. An exponential extrapolation of the data (dashed lines) suggests that the device switching on/off ratio will retain average of 6×10^5 for over 10 years.

Table SI. State-of-the-art memristive switches fabricated by solution-based inorganic materials using printing methods.

Structure	Printing technique	LRS resistance	Set/Reset voltage	Cycle endurance	Testing duration	Lifetime	Ref
Ag/MoS ₂ /Ag	Inkjet printing	$10^1 \Omega$	Set 1.75 V; Reset -0.7 V	300 cycles	10^7 s	10 years	Our work
Ag/ZnO/Ag	Electrohydrodynamic printing, spin coating	$10^6 \Omega$	Set 2 V; Reset -2 V	N/A	N/A	N/A	1
Ag/ZnO/Cu	Electrohydrodynamic Printing	$10^3 \Omega$	Set 1.25 V; Reset -1.25 V	500 cycles	N/A	N/A	2
Ag/TiO ₂ /Cu	Electrohydrodynamic Printing	$10^2 \Omega$	Set 0.7 V; Reset -0.7 V	N/A	N/A	N/A	3
Ag/ZnSnO ₃ /Ag	Screen Printing, Electrohydrodynamic Atomization	$10^7 \Omega$	Set 2 V; Reset -2 V	100 cycles	10^2 s	N/A	4
Au/Cu-SiO ₂ NWs/Cu	Aerosol-Jet Printing	$10^4 \Omega$	Set 3 V; Reset -3 V	10^4 cycles	10^6 s	10 years	5
Ag/a-TiO ₂ /Ag	Inkjet Printing	$10^7 \Omega$	Set 10 V; Reset -10 V	10^3 cycles	100 s	N/A	6
Ag/ZnO/Ag	Electrohydrodynamic Printing	$10^2 \Omega$	Set 3.75 V; Reset -3.75 V	10^3 cycles	N/A	N/A	7
Ag/ZrO ₂ /Ag	Electrospray Deposition, Electrohydrodynamic Printing	$10^2 \Omega$	Set 3.8 V; Reset -2.6 V	N/A	N/A	N/A	8
Ag/h-BN/Ag	Inkjet Printing	$10^5 \Omega$	Set 2 V; Reset -1 V	10^5 cycles	10^4 s	N/A	9

Ag/MoS ₂ /Ag	Aerosol-Jet Printing	10 ¹ Ω	Set 0.18 V - 0.30 V, Reset -0.1 V	100 cycles	10 ² s	N/A	10
Ag/WSe ₂ /Ag	Aerosol-Jet Printing, Pneumatic atomizer	10 ⁵ Ω	Set 0.7 V; Reset -0.25 V	N/A	N/A	N/A	11
Ag/TiO ₂ /Carbon	Screen Printing, Inkjet Printing	10 ³ Ω	Set 1 V; Reset -3 V	100 cycles	N/A	N/A	12
Ag/ZrO ₂ /Ag	Electrohydrodynamic Printing	10 ⁵ Ω	Set 3 V; Reset -3 V	100 cycles	N/A	N/A	13
Ag/Cr-N-doped TiO ₂ /Ag	Reverse offset printing, EHD Printing, EHDA Printing	10 ⁴ Ω	Set 1 V; Reset -1 V	500 cycles	5000 s	N/A	14
Ag/MoS ₂ /Gr	Inkjet printing	10 ⁴ Ω	Set 2 V; Reset -0.24 V	100 cycles	10 ² s	10 years	15

Q2: Figure 1 shows the changes in switch resistance after different cycles. It can be seen that the voltage of the resistance change is not stable. What is the reason for this?

Response from the authors: The changes in switch resistance after different cycles do not imply that the device is not stable. The switching of all types of non-volatile switches (also known as memristive switching devices, memristors) is a statistical process. It is related to the fact that the whisker growth depends a lot on the nucleation and local conditions for the growth of the conductive filaments. Since the local conditions change from cycle to cycle, it is natural to expect certain level of variations in the switching voltages and resistances of the ON and OFF states [Sun W, et al. "Understanding memristive switching via in situ characterization and device modeling." *Nature Communications* 10.1 (2019): 3453.] and [Feng X, et al. "A fully printed flexible MoS₂ memristive artificial synapse with femtojoule switching energy." *Advanced Electronic Materials* 5.12 (2019): 1900740.]. It is worth to notice that this phenomenon exists in all memristive switching devices, including those fabricated by CVD 2D materials [Pazos S, et al. "Memristive circuits based on multilayer hexagonal boron nitride for millimetre-wave radiofrequency applications." *Nature Electronics* (2024): 1-10.] and [Teja Nibhanupudi S. S., et al. "Ultra-fast switching memristors based on two-dimensional materials." *Nature Communications* 15.1 (2024): 2334.] and mechanical exfoliated 2D materials [Wang M, et al. "Robust memristors based on layered two-dimensional materials." *Nature Electronics* 1.2 (2018): 130-136.].

Q3: Is the process of resistance change related to the current carrying capacity of the sample?

Response from the authors: No, it is not. The statistical process of the formation of conductive filaments (which is the origin of the switching mechanism) is the cause of resistance change as explained in answering Q2 [Sun W, et al. "Understanding memristive switching via in situ characterization and device modeling." *Nature Communications* 10.1 (2019): 3453.] and [Feng X, et al. "A fully printed flexible MoS₂ memristive artificial synapse with femtojoule switching energy." *Advanced Electronic Materials* 5.12 (2019): 1900740.]. It should be noted that once the switch is on, the current can increase exponentially. This is why the current limit must be set for any non-volatile (or memristive) switches - without current limit, the device will burnout quickly. Such protection can be easily realized by using a series resistor in the current path. In our tests, 11 mA limit was applied. It is well known for such non-volatile switches that the higher the current limit, the lower the

resistance will be. It is however not the process of resistance change related to the current carrying capacity.

Q4: Is there an intermediate state related to voltage?

Response from the authors: No, there is no. We have applied different current limits in the I-V tests of our devices. The switching characteristic remain stable and non-volatile. The switching mode is stable, with all tests showing a fast switch from HRS to LRS and vice vice, providing no evidence of any intermediate state related to voltage. It is noted that a large current limit during the setting process produces lower LRS resistance as shown in **Figure R1**. With a reading voltage of 10 mV after setting the device to different current limits, we achieved multiple resistance states with a wide tunability range from $10\ \Omega$ to $10^4\ \Omega$, showing great potential for enabling multi-level RF switching, data storage and multi-state neuromorphic computing.

Figure R1. Multiple states of resistance recorded with various current limits. The reading voltage is 10 mV.

About the function of the metasurface, the coding sequence ‘111’ and ‘000’ should generate the same reflection angle because $d\phi(x)/dx$ are same. The authors do not give me a reasonable explanation. The size of the device is too small, the edge of device may cause extensive diffraction. Therefore, I strongly recommend the authors to fabricate a device with larger size (more cells) to get the demonstration again.

Response from the authors: We agree with the Reviewer - $d\phi(x)/dx$ are the same for ‘111’ and ‘000’, i.e., there is no phase difference between adjacent cells that have the same code. However, the

reflections generated by these two-coding sequence should have some differences, as shown in **Figure R2**. There are noticeable differences in the side lobes although the primary lobes for both '000' and '111' coding sequences remain rather similar. These differences are attributed to the fundamental structural differences in their respective unit cells. The '0' code represents a configuration where the two components of the unit cell are disconnected, whereas the '1' code indicates that the two components are connected. These variations in geometry result in different surface current distributions, as shown in **Figure R3**, generating different reflections (both magnitudes and phases). This phenomenon is consistent with the observations reported by Cui et al. in their study of coding, digital, and programmable metamaterials [Cui T, *et al.* "Coding metamaterials, digital metamaterials and programmable metamaterials." *Light Sci Appl* 3, e218 (2014)].

Figure R2. Simulated reflected far-field patterns (magnitude) of the non-volatile reconfigurable metasurface (6×6) at 3.54 GHz, with coding sequence (a) 000, and (b) 111. The incident wave angle is -50° .

Figure R3. Surface current density distributions at 3.54 GHz on the top layer of the metasurface for an incident wave ($\theta = -50^\circ$). (a) coding '0', (b) coding '1'.

We have fabricated a larger array (6×6) according to the Reviewer's suggestion and revised the text accordingly:

It can be observed that the coding sequence '001001' in Fig. 5(e) provides very different reflected beam patterns compared to those of the coding sequence '010010' and '001011' in Fig. 5(f) and (h), respectively. For the same incident angle, the main reflected beam in Fig. 5(f) points to 41° direction and in Fig. 5(h) point to 45° direction, whereas the main reflected beam in Fig. 5(e) directs to 52° , demonstrating the ability of the fully printed MoS₂ switch coded metasurface to designate a reflected wave to a desired direction. This is highly desirable for smart wireless environment where intelligent reflective surfaces (IRS) can be deployed to make the wireless signal path programmable. If the coding sequence becomes '101101', the reflected electromagnetic wave is spread out into many beams as illustrated in Fig. 5(g), revealing that the MoS₂ switch coded metasurface can manipulate the incident wave so to level the reflected wavefronts for RCS reduction.

Fig. 5 | Zero-static power MoS₂ switch coded RF/microwave reconfigurable metasurface and its performance. (a) Simulated phase responses of the metasurface for a normal electromagnetic wave incidence. (b) and (c) Photos of the whole fully printed 6×6 zero-static power MoS₂ switch coded RF/microwave reconfigurable graphene metasurface and its unit cell, respectively. (d) Photo of the setup in the anechoic chamber. Measured far field patterns with coding sequence of (e) 001001, (f) 010010, (g) 101101, and (h) 001011 with an oblique incident angle of -50° .